# Uncertainty Representations in State-Space Layers for Deep Reinforcement Learning under Partial Observability

**Carlos E. Luis** *carlos@robot-learning.de*
*Bosch Corporate Research*
*Intelligent Autonomous Systems Group, Technical University Darmstadt*

**Alessandro G. Bottero** *alessandrogiacomo.bottero@bosch.com*
*Bosch Corporate Research*
*Intelligent Autonomous Systems Group, Technical University Darmstadt*

**Julia Vinogradska** *julia.vinogradska@bosch.com*
*Bosch Corporate Research*

**Felix Berkenkamp** *felix.berkenkamp@bosch.com*
*Bosch Corporate Research*

**Jan Peters** *jan.peters@tu-darmstadt.de*
*Intelligent Autonomous Systems Group, Technical University Darmstadt*
*German Research Center for AI (DFKI)*
*Hessian.AI*
*Centre for Cognitive Science*

**Reviewed on OpenReview:** *https://openreview.net/forum?id=rfPnsOWJyg*

## Abstract

Optimal decision-making under partial observability requires reasoning about the uncertainty of the environment's hidden state. However, most reinforcement learning architectures handle partial observability with sequence models that have no internal mechanism to incorporate uncertainty in their hidden state representation, such as recurrent neural networks, deterministic state-space models and transformers. Inspired by advances in probabilistic world models for reinforcement learning, we propose a standalone Kalman filter layer that performs closed-form Gaussian inference in linear state-space models and train it end-to-end within a model-free architecture to maximize returns. Similar to efficient linear recurrent layers, the Kalman filter layer processes sequential data using a parallel scan, which scales logarithmically with the sequence length. By design, Kalman filter layers are a drop-in replacement for other recurrent layers in standard model-free architectures, but importantly they include an explicit mechanism for probabilistic filtering of the latent state representation. Experiments in a wide variety of tasks with partial observability show that Kalman filter layers excel in problems where uncertainty reasoning is key for decision-making, outperforming other stateful models.

## 1 Introduction

The classical reinforcement learning (RL) formulation tackles optimal decision-making in a fully observable Markov decision process (MDP) (Sutton and Barto, 2018). However, many real-world problems are *partially* observable, since we only have access to observations that hide information about the state, e.g., due to noisy measurements. Learning in partially observable MDPs (POMDPs) is statistically and computationally intractable in general (Papadimitriou and Tsitsiklis, 1987), but in many practical scenarios it is theoretically viable (Liu et al., 2022) and has lead to successful applications in complex domains like robotics (Zhu et al.,

2017), poker (Brown and Sandholm, 2019), real-time strategy games (Vinyals et al., 2019) and recommendation systems (Li et al., 2010).

Practical algorithms for RL in POMDPs employ sequence models that encode the history of observations and actions into a latent state representation amenable for policy optimization. Besides extracting task-relevant information from the history, probabilistic inference over the latent state is also crucial under partial observability (Kaelbling et al., 1998). As a motivating example, consider an AI chatbot that gives restaurant recommendations to users. Since the user's taste (i.e., the state) is unknown, the agent must ask questions before ultimately making its recommendation. Reasoning over the latent state uncertainty is crucial to decide whether to continue probing the user or end the interaction with a final recommendation. An optimal agent would gather enough information to recommend a restaurant with a high likelihood of user satisfaction. In Section 5.2, we evaluate performance of our proposed approach in a simplified version of this problem.

A standard recipe for model-free RL in POMDPs is to combine a sequence model (e.g., LSTM (Hochreiter and Schmidhuber, 1997), GRU (Cho et al., 2014)) with a policy optimizer (e.g., PPO (Schulman et al., 2017) or SAC (Haarnoja et al., 2018)), which has shown strong performance in a wide variety of POMDPs (Ni et al., 2022). More recently, transformers (Vaswani et al., 2017) have also been adopted as sequence models in RL showing improved memory capabilities (Ni et al., 2023). However, their inference runtime scales quadratically with the sequence length, which makes them unsuitable for online learning in physical systems (Parisotto and Salakhutdinov, 2020). Instead, recent deterministic state-space models (SSMs) (Gu et al., 2022a; Smith et al., 2023; Gu and Dao, 2023) maintain the constant-time inference of stateful models, while achieving logarithmic runtime during training thanks to efficient parallel scans (Smith et al., 2023). Moreover, SSMs have shown improved long-term memory, in-context learning and generalization in RL (Lu et al., 2023). Yet, in problems where reasoning over latent state uncertainty is crucial, it remains unclear whether such methods can learn the required probabilistic inference mechanisms for decision making. The core objective of this work is to study the role of *explicit* probabilistic inference within a model-free RL architecture for POMDPs.

While model-free architectures focus on *deterministic* sequence models, in model-based RL *probabilistic* sequence models are a widespread tool to model uncertainty in environment dynamics (Watter et al., 2015; Hafner et al., 2019; 2020; Becker and Neumann, 2022). Considering these two sequence modelling approaches, we concretely investigate the following questions:

> *Can we leverage the same inference methods developed for model-based RL as general-purpose sequence models in model-free architectures? If so, does it bring any benefits compared to deterministic models?*

Our core hypothesis is that explicit probabilistic inference in sequence models may serve as an inductive bias to learn in tasks where uncertainty over the latent state is crucial for decision making, as our motivating example on the restaurant recommendation chatbot.

**Our Contributions.** Inspired by the simple inference scheme in the Recurrent Kalman Network (RKN) (Becker et al., 2019) architecture for world models, we embed closed-form Gaussian inference in linear SSMs as a standalone recurrent layer — denoted a Kalman filter (KF) layer — and train it end-to-end within a model-free architecture (Ni et al., 2022) to maximize returns. Since our KF layers are designed to be a drop-in replacement for standard recurrent layers, they can also be stacked together and combined with other components (e.g., residual connections, normalization, etc.) to build more complex sequence models. Similar to Becker et al. (2024), we leverage the associative property of the Kalman filter operations for efficient training of KF layers via parallel scans, which scale logarithmically with the sequence length provided sufficient parallel GPU cores.

We systematically evaluate our research questions across a variety of POMDPs that probe distinct capabilities, such as uncertainty reasoning, adaptation, generalization and filtering of noisy observations. We benchmark the performance of KF layers against a wide range of baselines from prior work, including GRUs, transformers, and deterministic SSMs, all embedded in the same model-free architecture. To ensure fairness in our comparisons, we meticulously control for confounding factors such as parameter count, training procedure and hyperparameters. By holding all aspects of the architecture and training constant, we isolate the impact of each sequence model in the overall performance, providing clear insights into their relative effectiveness. Through these experiments, we demonstrate that KF layers can be trained effectively end-to-end on model-free

objectives, excelling in tasks where probabilistic inference is key for decision-making and showing significant improvements over deterministic stateful models.

## 2 Related Work

**RL architectures for POMDPs.** Partial observability in RL tasks requires agents to maintain *memory* of past interactions. Some approaches incorporate memory systems inspired by principles of human psychology, such as reward-based learning (Fortunato et al., 2019), or rely on mechanisms like context-dependent retrieval (Oh et al., 2016). A more widespread solution involves *sequence models*, also referred to as *history encoders* (Ni et al., 2024), which encode past observations and actions into a state representation useful for RL. These models have been used to augment policies (Wierstra et al., 2007), value functions (Schmidhuber, 1990; Bakker, 2001) and world models (Schmidhuber, 1991; Becker et al., 2019; Shaj et al., 2021a;b; 2023). This enables RL algorithms, such as DQN (Hausknecht and Stone, 2015), SAC (Ni et al., 2022), PPO (Kostrikov, 2018; Ni et al., 2023; Lu et al., 2023), DPG (Heess et al., 2015) and Dyna (Hafner et al., 2020; Becker and Neumann, 2022) to handle partial observability. In this work, we adopt an off-policy model-free architecture similar to Ni et al. (2022), leveraging its strong performance in various POMDPs.

**Sequence models in RL.** Frame-stacking was one of the earliest methods used in RL to capture temporal context by concatenating consecutive observations (Lin and Mitchell, 1993). It remains a common tool for conveying velocity information from image-based observations, such as in the Atari benchmark (Bellemare et al., 2013; Mnih et al., 2013). However, frame-stacking fails to model long-range dependencies in more complex POMDPs due to its fixed and shallow representation of temporal relationships. To address this limitation, stateful recurrent models became the dominant approach for extracting relevant information from arbitrarily long contexts. Examples include RNNs (Lin and Mitchell, 1993; Schmidhuber, 1990), LSTMs (Bakker, 2001) and GRUs (Kostrikov, 2018). More recently, the transformer architecture (Vaswani et al., 2017) has shown promise in improving the long-term memory in RL agents (Ni et al., 2023). However, while transformers excel at modeling long-range dependencies, their slow inference and large memory footprint reduce their practicality for real-time control tasks, where efficiency is critical (Parisotto and Salakhutdinov, 2020). These challenges emphasize the need for more efficient sequence models that balance representational power with computational feasibility.

**Deterministic SSMs.** State-space models are of particular interest to the RL community due to their computational efficiency compared to traditional sequence models like RNNs and transformers. They maintain the fast inference of RNNs, but scale logarithmically (rather than linearly) with the sequence length during training (Smith et al., 2023). Moreover, they also circumvent vanishing/exploding gradients with proper initialization (Gu et al., 2020) and match (or even exceed) the performance of transformers in long-range sequence modelling tasks (Lu et al., 2023). In particular, *structured* state space models such as S4 (Gu et al., 2022a), S5 (Smith et al., 2023) and S6 / Mamba (Gu and Dao, 2023) have emerged as a strong competitor to transformers in general sequence modelling problems like language (Fu et al., 2023), audio (Goel et al., 2022) and video (Nguyen et al., 2022). The adoption of these models in RL is still in its infancy, however. For instance, Morad et al. (2023) report bad performance of a variant of S4 (Gu et al., 2022b) in various POMDPs, while Lu et al. (2023) show that combining S5 with PPO yields strong results in long-term memory and in-context learning. These mixed findings suggest that the performance of deterministic SSMs in RL is sensitive to implementation details and possibly environment-dependent. Furthermore, it remains unclear how these models perform in tasks where uncertainty in the latent state is critical for decision-making, as they lack explicit probabilistic inference mechanisms. We hypothesize that probabilistic inference is vital to handle such problems.

**Probabilistic SSMs.** Probabilistic SSMs are a common tool in model-based RL to train both discriminative (Haarnoja et al., 2016) and generative (Ha and Schmidhuber, 2018) models of the environment, often referred to as world models. These models are trained to capture the environment's dynamics, which can then be used for: (i) planning (Hafner et al., 2019) or policy optimization (Becker and Neumann, 2022; Hafner et al., 2020) via latent imagination (i.e., generating imaginary policy rollouts auto-regressively), or (ii) policy optimization on the learned latent representation (Becker et al., 2024). A prominent approach is the Recurrent State Space Model (RSSM) proposed by Hafner et al. (2019), which divides the latent state into deterministic and

stochastic components and uses a GRU to propagate the deterministic part forward. More recently, GRUs have been replaced by transformers (Chen et al., 2021) and S4 (Samsami et al., 2024) models, albeit in a simplified inference scheme that conditions only on the current observation rather than the history, possibly for computational efficiency. A common objective function for training these probabilistic SSMs is the evidence lower bound (ELBO), which provides a lower bound on the log-likelihood of the environment's data. This ensures that generative models produce plausible trajectories given an action sequence, typically optimized with variational autoencoders (Kingma and Welling, 2014). These autoencoders shape a low-dimensional latent representation to capture salient features of the environment's data generation process. The world model objective can be viewed as an auxiliary loss that helps the sequence model learn a useful representation for control tasks, which has shown improved sample efficiency in problems with complex, high-dimensional observations like images (Hafner et al., 2023). Other approaches, such as contrastive learning (Laskin et al., 2020), similarly propose a proxy objective that shapes the learned representation to maximize agreement between augmented views of the environment. These auxiliary losses has benefits and drawbacks: they provide a strong learning signal for representation learning, even in the absence of a reward signal, but they introduce complexity in training and can interfere with the RL objective, known as the objective mismatch problem (Lambert et al., 2020). While these tradeoffs warrant research on their own, the purpose of this work is to bring understanding in the role of probabilistic inference in model-free RL architectures *without* auxiliary objectives.

**Kalman filters.** A particular class of probabilistic SSMs of relevance to this work are Kalman filters (Kalman, 1960), which perform optimal inference in *linear* SSMs under a Gaussian noise assumption. Since then, Kalman filters have been theoretically extended to handle non-linear dynamics (Serra, 2018) and also widely adopted in a range of science and engineering fields (Auger et al., 2013), including robotics (Urrea and Agramonte, 2021), vision (Chen, 2012), signal processing and sensor fusion (Khaleghi et al., 2013). They also have a rich history within the machine learning community, particularly in early applications for time-series forecasting (Shumway and Stoffer, 1982). While the linear-Gaussian assumption in standard Kalman filtering is restrictive for the high-dimensional data often found in machine learning applications (Murphy, 2012; Bishop, 2006), several extensions have been proposed. One class of approaches circumvent these limitations by modelling non-linear state transitions with neural networks (Krishnan et al., 2015; 2017) and performing approximate inference via stochastic gradient variational Bayes (Kingma and Welling, 2014). Alternatively, other approaches simplify the inference problem by embedding (locally) linear-Gaussian SSMs in learned latent spaces (Watter et al., 2015; Karl et al., 2017; Klushyn et al., 2021), enabling exact Kalman filtering and yielding better performance than methods with more complex dynamics but poor approximate inference (Fraccaro et al., 2017). However, the use of full transition and covariance matrices limits the practical dimensionality of the latent space and the expressivity of the models. In contrast, we adopt a simpler parameterization of the linear-Gaussian SSM, using diagonal matrices and covariances, which significantly reduces the computational burden of Kalman filtering. This approach scales to higher-dimensional latent spaces, enables logarithmic scaling (in the sequence length) of the Kalman filter equations (Sarkka and Garcia-Fernandez, 2021; Becker et al., 2024) and preserves the expressivity of the models by offloading representational power to other components of the architecture, such as encoders and decoders (Haarnoja et al., 2016; Becker et al., 2019).

**Kalman filters in RL.** Kalman filters have been extensively used in RL as discriminative (Haarnoja et al., 2016; Becker et al., 2019; Shaj et al., 2021a;b; 2023) or generative (Watter et al., 2015; Becker and Neumann, 2022; Becker et al., 2024) world models. The former are trained on regression losses to obtain accurate *predictions*, while the latter are trained with variational inference for temporally-consistent *generation*. In the model-free architecture considered in this work, only discriminative sequence models can be integrated without altering the training procedure; generative models would require auxiliary loss functions, which would modify the training process and introduce potential confounding factors that are not part of our experimental design. Closest to our approach is the Recurrent Kalman Network (RKN) (Becker et al., 2019), an encoder-decoder architecture that employs Kalman filtering using locally linear models and structured (non-diagonal) covariance matrices. Follow-up work has extended the RKN framework in various ways: Shaj et al. (2021a) include action conditioning, Shaj et al. (2021b) consider a multi-task setting with hidden task parameters and Shaj et al. (2023) propose a hierarchical, multi-timescale architecture. While these approaches train the latent representation to capture the environment's dynamics, our work instead focuses on training a

similar Kalman filter end-to-end with the RL objective — return maximization — such that the latent space is shaped specifically for control rather than prediction.

## 3 Background

In this section, we provide the relevant background and introduce core notation used throughout the paper. We use bold upper case letters ($\mathbf{A}$) to denote matrices and calligraphic letters ($\mathcal{X}$) to denote sets. The notation $\text{diag}(\mathbf{A})$ refers to a vector containing the diagonal elements for a square matrix $\mathbf{A}$ and $\mathcal{P}(\mathcal{X})$ refers to the space of probability distributions over $\mathcal{X}$.

### 3.1 Reinforcement Learning in Partially Observable Markov Decision Processes

We consider an agent that acts in a finite-horizon partially observable Markov decision process (POMDP) $\mathcal{M} = \{\mathcal{S}, \mathcal{A}, \mathcal{O}, T, p, O, r, \gamma\}$ with state space $\mathcal{S}$, action space $\mathcal{A}$, observation space $\mathcal{O}$, horizon $T \in \mathbb{N}$, transition function $p : \mathcal{S} \times \mathcal{A} \to \mathcal{P}(\mathcal{S})$ that maps states and actions to a probability distribution over $\mathcal{S}$, an emission function $O : \mathcal{S} \to \mathcal{P}(\mathcal{O})$ that maps states to a probability distribution over observations, a reward function $r : \mathcal{S} \times \mathcal{A} \to \mathbb{R}$, and a discount factor $\gamma \in [0, 1)$.

At time step $t$ of an episode in $\mathcal{M}$, the agent observes $o_t \sim O(\cdot \mid s_t)$ and selects an action $a_t \in \mathcal{A}$ based on the observed history $h_{:t} = (o_{:t}, a_{:t-1}) \in \mathcal{H}_t$, then receives a reward $r_t = r(s_t, a_t)$ and the next observation $o_{t+1} \sim O(\cdot \mid s_{t+1})$ with $s_{t+1} \sim p(\cdot \mid s_t, a_t)$.

We adopt the general setting by Ni et al. (2023; 2024), where the RL agent is equipped with: ($i$) a stochastic policy $\pi : \mathcal{H}_t \to \mathcal{P}(\mathcal{A})$ that maps from observed history to distribution over actions, and ($ii$) a value function $Q^\pi : \mathcal{H}_t \times \mathcal{A} \to \mathbb{R}$ that maps from history and action to the expected return under the policy, defined as $Q^\pi(h_{:t}, a_t) = \mathbb{E}_\pi \left[ \sum_{h=t}^T \gamma^{h-t} r_t \mid h_{:t}, a_t \right]$. The objective of the agent is to find the optimal policy that maximizes the value starting from some initial state $s_0$, $\pi^\star = \text{argmax}_\pi \mathbb{E}_\pi \left[ \sum_{t=0}^{T-1} \gamma^t r_t \mid s_0 \right]$.

### 3.2 History Representations

A weakness of the general formulation of RL in POMDPs is the dependence of both the policy and the value function on the ever-growing history. Instead, practical algorithms fight this curse of dimensionality by *compressing* the history into a compact representation. Ni et al. (2024) propose to learn such representations via *history encoders*, defined by a mapping $\phi : \mathcal{H}_t \to \mathcal{Z}$ from observed history to some latent representation $z_t := \phi(h_{:t}) \in \mathcal{Z}$. With slight abuse of notation, we denote $\pi(a_t \mid z_t)$ and $Q^\pi(z_t, a_t)$ as the policy and values under this latent representation, respectively.

### 3.3 Probabilistic Inference on Linear SSMs

We consider time-varying, discrete, linear-Gaussian SSMs defined by

$$x_t = \mathbf{A}_t x_{t-1} + \mathbf{B}_t u_{t-1} + \varepsilon_t, \quad y_t = \mathbf{C}_t x_t + \mathbf{D}_t u_{t-1} + \nu_t, \tag{1}$$

where $t > 0 \in \mathbb{N}$, $x_t \in \mathbb{R}^N$ is the hidden or latent state, $u_t \in \mathbb{R}^P$ is the input, $y_t \in \mathbb{R}^M$ is the output, $(\mathbf{A}_t, \mathbf{B}_t, \mathbf{C}_t, \mathbf{D}_t)$ are matrices of appropriate size, $\varepsilon_t \sim \mathcal{N}(0, \boldsymbol{\Sigma}_t^{\text{p}})$ and $\nu_t \sim \mathcal{N}(0, \boldsymbol{\Sigma}_t^{\text{o}})$ are zero-mean process and observation noise variables with their covariance matrices $\boldsymbol{\Sigma}_t^{\text{p}}$ and $\boldsymbol{\Sigma}_t^{\text{o}}$, respectively. Without loss of generality and as it is common in linear SSMs, we set $\mathbf{D}_t \equiv \mathbf{0}$. The latent state probabilistic model is then $p(x_t \mid x_{t-1}, u_{t-1}) = \mathcal{N}(\mathbf{A}_t x_{t-1} + \mathbf{B}_t u_{t-1}, \boldsymbol{\Sigma}_t^{\text{p}})$ and the observation model is $p(y_t \mid x_t) = \mathcal{N}(\mathbf{C}_t x_t, \boldsymbol{\Sigma}_t^{\text{o}})$. Inference in such a model has a closed-form solution, which is equivalent to the well-studied Kalman filter (Kalman, 1960).

**Predict.** The first stage of the Kalman filter propagates forward the *posterior* belief of the latent state at step $t-1$, given by $\mathcal{N}(x_{t-1}^+, \boldsymbol{\Sigma}_{t-1}^+)$, to obtain a *prior* belief at step $t$, $\mathcal{N}(x_t^-, \boldsymbol{\Sigma}_t^-)$, given by

$$x_t^- = \mathbf{A}_t x_{t-1}^+ + \mathbf{B}_t u_{t-1}, \quad \boldsymbol{\Sigma}_t^- = \mathbf{A}_t \boldsymbol{\Sigma}_{t-1}^+ \mathbf{A}_t^\top + \boldsymbol{\Sigma}_t^{\text{p}}. \tag{2}$$

**Update.** The second stage updates the prior belief at step $t$ given some observation $w_t$, to obtain the posterior $p(x_t \mid x_{t-1}, w_t) = \mathcal{N}(x_t^+, \Sigma_t^+)$ given by

$$x_t^+ = x_t^- + \mathbf{K}_t(w_t - \mathbf{C}_t x_t^-), \quad \mathbf{\Sigma}_t^+ = (\mathbf{I} - \mathbf{K}_t \mathbf{C}_t)\mathbf{\Sigma}_t^-, \tag{3}$$

where $\mathbf{K}_t = \mathbf{\Sigma}_t^- \mathbf{C}_t^\top (\mathbf{C}_t \mathbf{\Sigma}_t^- \mathbf{C}_t^\top + \mathbf{\Sigma}_t^o)^{-1}$ is known as the Kalman gain. The predict and update steps are interleaved to process sequences of input and observations $\{u_t, w_t\}_{t=0}^{K-1}$ of length $K$, starting from some initial belief $\mathcal{N}(x_{-1}^+, \mathbf{\Sigma}_{-1}^+)$.

### 3.4 Simplifying Assumptions

The Kalman filter predict and update equations from (2) and (3) involve expensive matrix multiplication and inversion, which scales poorly with the latent state dimension $N$. In this section we propose several simplifications, both for easier implementation but also for better scalability.

**Time-invariance.** Prior work proposed time-varying SSMs via state-dependent (Becker et al., 2019) or input-dependent (Gu and Dao, 2023) matrices. Instead, in this work we propose using simple time-invariant matrices, as similarly done in prior deterministic SSMs (Gu et al., 2022a; Smith et al., 2023). First, state-dependent matrices is well motivated by local linearization of dynamics, but they are incompatible with efficient parallel scan routines, as they break the associative property of the Kalman filter equations. Second, while input-dependent matrices excel in associative recall problems where input-selectivity is necessary, Kalman filters provide similar selection mechanisms via its posterior update (see Section 4.2), without the need for time-varying matrices. Concerning time-invariant process noise, such simplification reduces the expressivity of the model. However, our initial experiments with input-dependent process noise showed *worse* performance in RL than its time-invariant alternative (see Section 4.2 and Appendix E).

**Diagonal matrices.** In order to scale to higher-dimensional latent spaces, prior work in both deterministic and probabilistic SSMs consider *structured* SSMs. This simply means special structure is imposed into the learnable matrices $(\mathbf{A}, \mathbf{B}, \mathbf{C})$. In particular, we consider a diagonal structure with the HiPPO initialization proposed in Gu et al. (2020), which induces stability in the recurrence for handling long sequences. In addition, we also consider: (i) diagonal process and observation noise covariances, (ii) $N = M = P$, which simplifies the implementation and (iii) identity emission matrices $\mathbf{C} = \mathbf{I}$, as proposed in (Becker and Neumann, 2022). Under this parameterization, the expensive Kalman filter equations reduce to element-wise operations:

$$x_t^- = \mathrm{diag}(\mathbf{A}) \odot x_{t-1}^+ + \mathrm{diag}(\mathbf{B}) \odot u_{t-1}, \quad \mathrm{diag}(\mathbf{\Sigma}_t^-) = \mathrm{diag}(\mathbf{A})^2 \odot \mathrm{diag}(\mathbf{\Sigma}_{t-1}^+) + \mathrm{diag}(\mathbf{\Sigma}_t^p), \tag{4}$$

$$x_t^+ = x_t^- + \mathrm{diag}(\mathbf{K}_t) \odot (w_t - x_t^-), \quad \mathrm{diag}(\mathbf{\Sigma}_t^+) = (\mathrm{diag}(\mathbf{I}) - \mathrm{diag}(\mathbf{K}_t)) \odot \mathrm{diag}(\mathbf{\Sigma}_t^-), \tag{5}$$

$$\mathrm{diag}(\mathbf{K}_t) = \mathrm{diag}(\mathbf{\Sigma}_t^-) \oslash (\mathrm{diag}(\mathbf{\Sigma}_t^-) + \mathrm{diag}(\mathbf{\Sigma}_t^o)), \tag{6}$$

where $\odot$ denoted element-wise vector product and $\oslash$ denotes element-wise vector division.

### 3.5 Parallel Scans

Efficient implementation of state-space models and Kalman filters employ *parallel scans* to achieve logarithmic runtime scaling with the sequence length (Smith et al., 2023; Sarkka and Garcia-Fernandez, 2021). Given a sequence of elements $(a_0, a_1, \ldots, a_{t-1})$ and an *associative*[1] binary operator $\bullet$, the parallel scan algorithm outputs all the prefix-sums $(a_0, a_0 \bullet a_1, \ldots, a_0 \bullet \ldots \bullet a_{t-1})$ in $\mathcal{O}(\log K)$ runtime, given sufficient parallel processors.

## 4 Method: Off-Policy Recurrent Actor-Critic with Kalman filter Layers

In this section, we describe our method that implements Kalman filtering as a recurrent layer within a standard actor-critic architecture.

---

[1] A binary operator $\bullet$ is associative if $(a \bullet b) \bullet c = a \bullet (b \bullet c)$ for any triplet of elements $(a, b, c)$

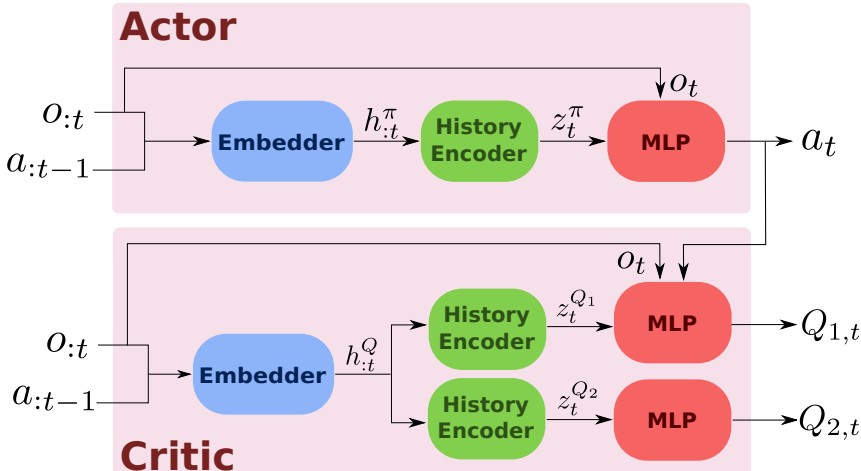

Figure 1: General Recurrent Actor-Critic (RAC) architecture. The components are trained end-to-end with the Soft Actor-Critic (SAC) loss function (Haarnoja et al., 2018). To handle discrete action spaces, we use the discrete version of SAC by Christodoulou (2019).

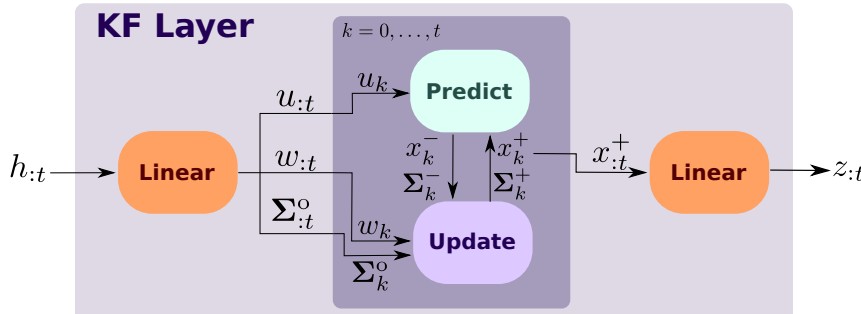

Figure 2: Our proposed Kalman filter layer to build history encoders. The KF layer receives a history sequence $h_{:t}$ and projects it into three separate signals in latent space: the input $u_{:t}$, the observation $w_{:t}$ and the observation noise (diagonal) covariance $\boldsymbol{\Sigma}^{\text{o}}_{:t}$. These sequences are processed using the standard Kalman filtering equations, which scale logarithmically with the sequence length using parallel scans. Lastly, the posterior mean latent state $x^{+}_{:t}$ is projected from the latent space back into the history space to obtain the compressed representation $z_{:t}$.

### 4.1 General Architecture

In Figure 1 we present our Recurrent Actor-Critic (RAC) architecture inspired by Ni et al. (2022), where we replace the RNN blocks with general history encoders. We will use this architecture in the following to test the capabilities of different history encoders in various POMDPs.

For both actor and critic, we embed the sequence of observations and actions into a single representation $h^{*}_{:t}$ which is then passed into the history encoders. We use a single linear layer as embedder, which we found worked as reliably as more complex non-linear embedders used in similar RAC architectures by Morad et al. (2023); Ni et al. (2022). We also include the skip connections from current observations and actions into the actor-critic heads, as proposed in previous memory-based architectures (Zintgraf et al., 2021; Ni et al., 2022).

### 4.2 Kalman Filter Layers

Our main hypothesis is that principled probabilistic filtering within history encoders boosts performance in POMDPs, especially those where reasoning about uncertainty is key for decision-making. To test this hypothesis, we introduce KF layers, as shown in Figure 2. The layer receives as input a history embedding

sequence $h_{:t}$ which is then projected into the input $u_{:t}$, observation $w_{:t}$ and observation noise $\boldsymbol{\Sigma}_{:t}^{\text{o}}$ sequences. These three signals serve as input to the standard KF predict-update equations (2) and (3), which output a posterior (filtered) latent state $x_{:t}^+$. Finally, the posterior sequence is projected back to the history embedding space to produce the compressed history representation $z_{:t}$.

**History encoders with KF layers.** Similar to recent SSM layers such as S5 (Smith et al., 2023) and S6 (Gu and Dao, 2023), these KF layers can be stacked and combined with other operations such as residual connections, gating mechanisms, convolutions and normalization to compose a history encoder block in the RAC architecture. In favor of simplicity, our history encoders are only composed of KF layers and (optionally) an RMS normalization (Zhang and Sennrich, 2019) output block for improved stability.

**Filtering as a gating mechanism.** We can draw interesting comparisons between KF layers and other recurrent layers from the perspective of gating mechanisms. It was shown in Theorem 1 of (Gu and Dao, 2023) that selective SSMs (S6) behave as generalized RNN gates through an input-dependent step size $\Delta$. In this case, the gate depends on the SSM input and controls how much the input influences the next hidden state. Similarly, as hinted by Becker et al. (2019), during the update step the Kalman gain is effectively an uncertainty-controlled gate depending on the observation noise which regulates how much the observation influences the posterior belief over the latent state. Our experiments in Section 5 shed some light on the strengths and weaknesses of these approaches for RL under partial observability.

**SSM Parameterization.** We follow the procedure in Gu and Dao (2023) and initialize the continuous-time system $(\tilde{\mathbf{A}}, \tilde{\mathbf{B}})$ with HiPPO matrices. The corresponding discrete-time system $(\mathbf{A}, \mathbf{B})$ is obtained via zero-order hold discretization with a learnable scalar step size $\Delta > 0$ (Smith et al., 2023).

**Design decisions.** We want to highlight two considerations that went into the design of our KF layers. First, we could generalize the architecture to support time-varying process noise by including one extra output channel (alongside the input, observation and observation noise channels) in the history linear projection. Conceptually, such an input-dependent process noise adds more flexibility to the gating mechanism implemented within the KF layer, which would be controlled both by the observation and the process noise signals. Second, we could include the posterior covariance $\boldsymbol{\Sigma}_{:t}^+$ as an additional feature for the output linear projection, alongside the posterior mean $x_{:t}^+$. We conduct an ablation study over these two choices in several continuous control tasks subject to observation noise and report the results in Appendix E. The best aggregated performance in this ablation was obtained with time-invariant process noise and only using the posterior mean as a feature for the output projection, which empirically justifies our final design.

## 4.3 Masked Associative Operators for Variable Sequence Lengths

In off-policy RAC architectures, the agent is typically trained with batches of (sub-)trajectories of possibly different length, sampled from an experience replay buffer. Thus, history encoders must be able to process batches of variable sequence length during training.

A common approach is to right-pad the batch of sequences up to a common length and ensure the model's output is independent of the padding values. For transformer models, this can be achieved by using the padding mask as a self-attention mask. For stateful models like RNNs and SSMs, it is imperative to also output the correct final latent state for each sequence in the batch. This typically requires a post-processing step that individually selects for each sequence in the batch the last state before padding. It turns out that for any recurrent model expressed with an associative operator (e.g., SSMs and KFs), we can obtain the correct final state from a batch of padded sequences *without* additional post-processing by using a parallel scan routine with a *Masked Associative Operator* (MAO).

**Definition 1** (Masked Associative Operator)**.** Let $\bullet$ be an associative operator acting on elements $e \in \mathcal{E}$, such that for any $a, b, c \in \mathcal{E}$, it holds that $(a \bullet b) \bullet c = a \bullet (b \bullet c)$. Then, the MAO associated with $\bullet$, denoted $\tilde{\bullet}$, acts on elements $\tilde{e} \in \mathcal{E} \times \{0, 1\} = (e, m)$, where $m \in \{0, 1\}$ is a binary mask. Then, for $\tilde{a} = (a, m_a)$ and $\tilde{b} = (b, m_b)$, we have:

$$\tilde{a} \mathbin{\tilde{\bullet}} \tilde{b} = \begin{cases} (a \bullet b, m_a) & \text{if} \quad m_b = 0 \\ \tilde{a} & \text{if} \quad m_b = 1 \end{cases} \tag{7}$$

In Appendix A, we show that any MAO is itself *associative* as long as we apply a right-padding mask[2], thus fulfilling the requirement for parallel scans. In practice, augmenting existing SSM and KF operators with their MAO counterpart is a minor code change. MAOs act as a pass-through of the hidden state when padding is applied, thus yielding the correct state at every time step of the padded sequence for each element of the batch without additional indexing or bookkeeping. Due to their pass-through nature, MAOs require strictly equal or less evaluations of the underlying associative operator, which may yield faster runtimes if the operator is expensive to evaluate and/or many elements of the input sequence are masked.

**MAOs for SSMs and KFs.** As a concrete example, the associative operators for SSMs and KFs involve matrix product and addition. A compute-efficient implementation of MAOs for such operators involves sparse matrix operations, where the sparsity is dictated by the padding mask. However, sparse matrix operations are only expected to yield better runtime than their dense counterparts for large matrices with sufficient levels of sparsity, which are not typical in our application. Thus, no speed-up is expected from using MAOs in the context of this work.

MAOs are similar to the custom operator proposed by Lu et al. (2023), but their effect is fundamentally different: Lu et al. (2023) considers on-policy RL, where the goal is to handle multi-episode sequences, thus their custom operator resets the hidden state at episode boundaries. Instead, in our off-policy RAC architecture, MAOs act as pass-through of the hidden state for padded inputs.

## 5 Experiments

In this section, we evaluate the RAC architecture under different history encoders in various POMDPs. Implementation details and hyperparameters are included in Appendices B and C, respectively.

### 5.1 Baselines

We consider the following implementation of history encoders within the RAC architecture.

`vSSM`. Vanilla, real-valued SSM with diagonal matrices. It is equivalent to a KF layer with infinite observation noise, i.e., the update step has no influence on the output. It can also be seen as a simplification of the S4D model (Gu et al., 2022b), where states are real-valued rather than complex (as in Mega (Ma et al., 2023), such that the recurrence can be interpreted as an exponential moving average) and the recurrence is implemented with a parallel scan rather than a convolution (as in (Smith et al., 2023)).

`vSSM+KF`. Probabilistic SSM via the KF layers described in Figure 2. While `vSSM` only *predicts* the next state (i.e., the prior in Kalman filtering), `vSSM+KF` additionally *filters* the predicted state conditioned on the latent observation. Therefore, `vSSM+KF` is equivalent to `vSSM` with the additional update step of the Kalman filter. Similarly, `vSSM` is equivalent to `vSSM+KF` with an infinite observation noise variance.

`vSSM+KF-u`. Equivalent to `vSSM+KF` *without* the input signal $u_{:t}$. It maintains the uncertainty-based gating from the KF layer, but looses flexibility in the KF predict step to influence the prior belief via the input.

**Mamba (Gu and Dao, 2023).** Selective state-space model with input-dependent state transition matrices.

**GRU (Cho et al., 2014).** Stateful model with a gating mechanism and non-linear state transitions.

**vTransformer (Vaswani et al., 2017).** Vanilla encoder-only transformer model with sinusoidal positional encoding and causal self-attention.

All SSM-based approaches are implemented using MAOs and parallel scans. Besides these memory-based agents, we include two additional memoryless agents that implement the same RAC architecure but without embedders or history encoders.

**Oracle.** It has access to the underlying state of the environment, effectively removing the partial observability aspect of the problem. This method should upper-bound the performance of history encoders.

---

[2]A right-padding mask is a sequence $\{m_0, m_1, \dots\}$ with $m_i \in \{0, 1\}$ such that if $m_i = 1$ then $m_j = 1$ for all $j > i$.

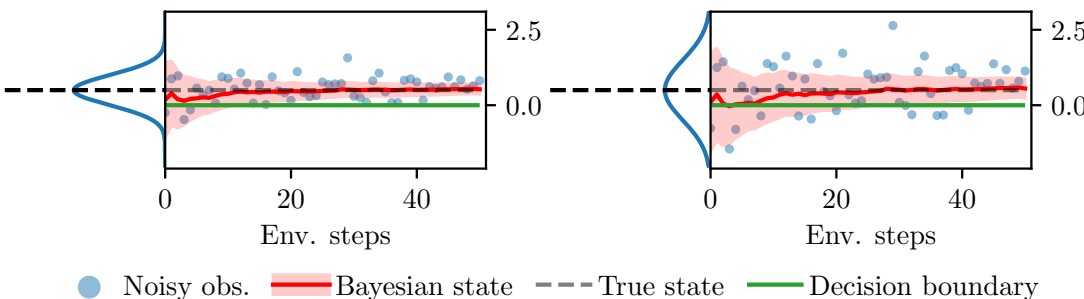

Figure 3: Two example episodes of the Best Arm Identification task of Section 5.2, with $\mu_b = 0.5$ and two different noise scales. **(Left)** Narrow noise distribution with $\sigma_b = 0.5$. **(Right)** Wide noise distribution with $\sigma_b = 1.0$. In red, we visualize the Bayesian posterior mean and $3\sigma$ confidence interval around $\mu_b$, obtained via Bayesian linear regression using all prior observations in the episode.

**Memoryless.** Unlike `Oracle`, it does not have access to the underlying state of the environment. This method should lower-bound the performance of history encoders.

All the baselines share a common codebase and hyperparameters. For all stateful models, we use the same latent state dimension $N$ such that parameter count falls within a 10% tolerance range except for `GRU`, which naturally has more parameters due to its gating mechanism (roughly 40% increase). For `vTransformer` we choose the dimension of the feed-forward blocks such that the total parameter count is also within 10% of the SSM methods. With this controlled experimental setup, we aim to evaluate strengths and weaknesses of the different mechanisms for sequence modelling (gating, input-selectivity, probabilistic filtering, self-attention) in a wide variety of partially observable environments.

## 5.2 Probabilistic Reasoning - Adaptation and Generalization

We evaluate probabilistic reasoning capabilities with a carefully designed POMDP that simplifies our motivating example from Section 1, where an AI chatbot probes a user in order to recommend a restaurant. Given noisy scalar observations sampled from a bandit with distribution $\mathcal{N}(\mu_b, \sigma_b)$, the task is to infer whether the mean $\mu_b$ lies above or below zero. At the start of each episode, $\mu_b$ and $\sigma_b$ (the latent parameters) are sampled from some given distribution. Then, at each step of an episode, the RL agent has three choices: (1) request a new observation from the bandit, which incurs a cost $\rho$, (2) decide the arm has mean above zero or (3) decide the arm has mean below zero, both of which immediately end the episode and provide a positive reward if the decision was correct, or a negative reward if the decision was incorrect. We set a maximum episode length of 1000 steps; if the agent does not issue a decision by then, it receives the negative reward. Example rollouts for this environment are provided in Figure 3. Given the Bayesian state from Figure 3, an optimal agent must strike a balance between requesting new information (which reduces uncertainty about the estimated mean) and minimizing costs. Effective history encoders for this problem should similarly produce a state representation that encodes uncertainty about the latent parameters.

We evaluate two core capabilities: adaptation and generalization. Intuitively, an optimal policy for this problem must be *adaptive* depending on the latent parameters. For example, if $\mu_b$ is close to zero the agent might need many observations to make an informed decision, whereas with a large $|\mu_b|$ the correct decision can be made with few observations. Moreover, we can also evaluate *generalization* of the learned policy by testing on latent parameters not seen during training. Our hypothesis is that an agent that learns proper probabilistic reasoning (e.g., Bayes' rule) should generalize reasonably well in this task.

We conduct experiments for all baselines under increasing cost $\rho$. Instead of providing the latent parameters directly to the `Oracle` baseline, we provide the Bayesian posterior mean and standard deviation around the latent parameter $\mu_b$, as shown in Figure 3. The agents are trained under the latent parameter distribution given by $\mu_b \sim \text{Unif}(-0.5, 0.5)$ and $\sigma_b \sim \text{Unif}(0.0, 2.0)$. We additionally evaluate out-of-distribution generalization by using $\sigma_b^{\text{OOD}} \sim \text{Unif}(2.0, 3.0)$, i.e., we test how the agent generalizes to bandits with higher variance. In Figure 4

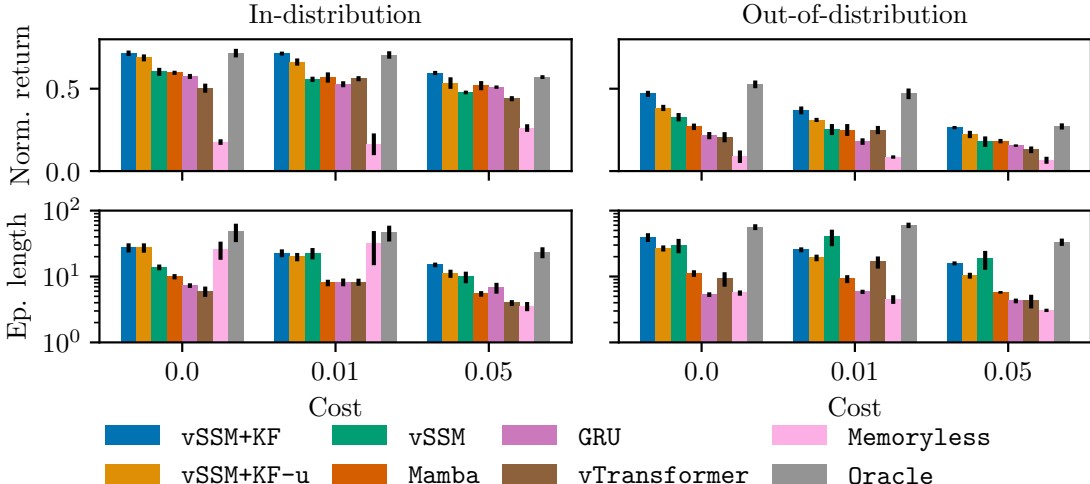

Figure 4: Performance of sequence models in the Best Arm Identification problem after 500K environment steps. We conduct experiments for increasing cost of requesting new observations and evaluate performance both in and out of distribution, averaged over 100 episodes, and report the mean and standard error over 5 random seeds. **(Top row)** Normalized return, obtained by dividing returns by the reward given after winning (10 in our case). **(Bottom row)** Length of episodes.

we report the normalized return and average episode length for both the training and out-of-distribution latent parameters. Full training curves are included in Appendix D. `vSSM+KF` achieves the highest return out of the memory-based agents, both in and out-of-distribution, while matching the performance of `Oracle` in-distribution. The better performance of `vSSM+KF` correlates with longer episodes: compared to the other baselines, `vSSM+KF` learns to request more observations in order to issue a more informed decision.

**`vSSM+KF` improves adaptation and generalization.** To gain further insights on the results, we do a post-training evaluation on a subset of the agents across the entire latent parameter space, as shown in Figure 5. `vSSM+KF` learns adaptation patterns similar to `Oracle`: the length of episodes increase as the noise scale $\sigma_b$ increases and decrease as $|\mu_b|$ increases, as it is intuitively expected. Such adaptation is less pronounced in `vSSM`, `vSSM+KF-u` and `Mamba`, where episodes are shorter and ultimately results in lower win rates. While `vSSM+KF` does not match the generalization performance of `Oracle`, it remains the best amongst the history encoder baselines. Given our controlled experimental setup, we attribute the enhanced adaptation and generalization of `vSSM+KF` to the internal probabilistic filtering implemented in the KF layer. Moreover, comparing `vSSM+KF` and `vSSM+KF-u` highlights that including the input signal in the KF layer leads to improved performance in this task.

**`vSSM+KF` can handle adversarial episodes.** In Figure 6 we compare latent space rollouts[3] from `vSSM+KF` and `vSSM` in an adversarial episode: $\mu_b$ is negative, but the first two observations are positive and of relatively large magnitude. After only four observations, `vSSM` is mislead by the positive observations and issues the wrong decision, as visualized in Figure 6 (middle) where we show the policy's output across latent space, overlaid with the rollout trajectory. Instead, `vSSM+KF` remains in the region where the policy requests more observations before it navigates to the correct region of latent space, as shown in Figure 6 (right). While this example was hand-picked, it is consistent with the adaptation patterns from Figure 5.

### 5.3 Probabilistic Filtering - Continuous Control under Observation Noise

In this experiment, we evaluate the ability to learn control policies subject to observation noise. Effective history encoders must learn to aggregate observations over multiple time steps to produce a filtered state

---

[3]We use a latent state dimension $N = 2$ in order to plot the policy decision boundary in latent space. This results in slightly worse performance than the results reported in Figure 4, where we use $N = 128$.

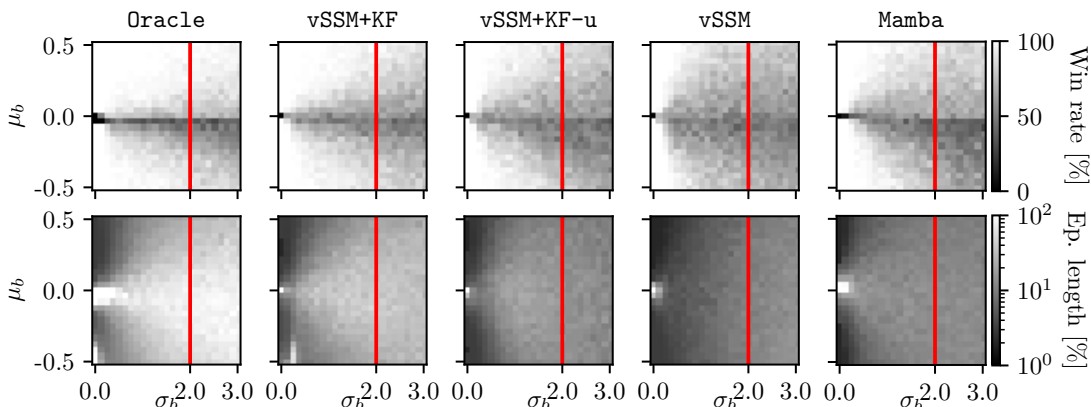

Figure 5: Performance heatmap on Best Arm Identification problem ($\rho = 0$). We generate a grid of noise parameters ($\mu_b, \sigma_b$) for a total of 625 unique combinations. The red vertical line separates training (to the left) from out-of-distribution (to the right) latent parameters. For each pair of latent parameters, we evaluate performance on five independently trained agents over 100 episodes and report the average win rate and episode lengths.

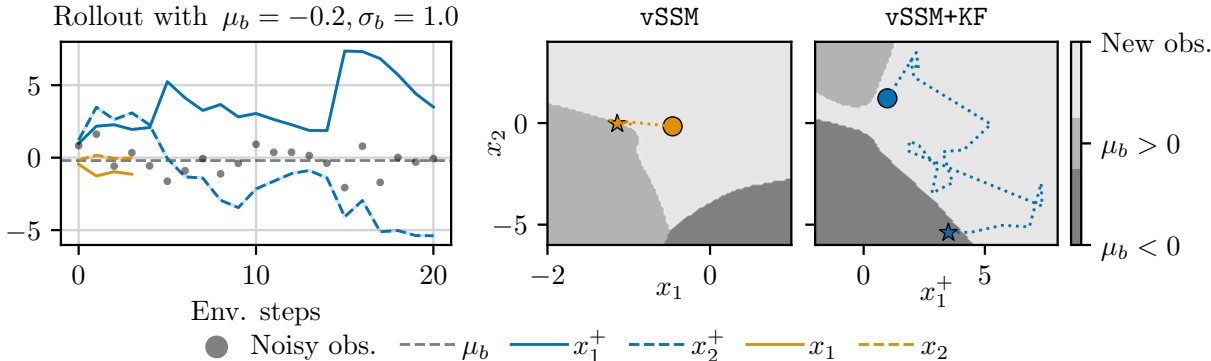

Figure 6: Latent space rollouts in adversarial Best Arm Identification episode. **(Left)** Rollout in latent space ($N = 2$) for vSSM+KF and vSSM after training. **(Middle-Right)** Policy decision boundaries overlaid with the latent space trajectory. Circles and stars denote the beginning and end of trajectories, respectively.

representation amenable for control. Our hypothesis is that internal probabilistic filtering provides an inductive bias for learning such a filtered representation. To test our hypothesis, we conduct evaluations across nine environments from the DeepMind Control (DMC) suite (Tunyasuvunakool et al., 2020) with zero-mean Gaussian noise added to the observations, as done by Becker and Neumann (2022); Becker et al. (2024). We present aggregated performance in Figure 7 following the recommendations from (Agarwal et al., 2021). Detailed training curves are included in Appendix F. We now discuss the main insights from this experiment.

**vSSM+KF improves performance of stateful models.** The KF layer is the only evaluated add-on for stateful models that significantly improves performance over the baseline model vSSM. This suggests that the uncertainty-based gating in Kalman filters is more effective at handling noisy data compared to the gating mechanism implemented by GRU and Mamba. This observation matches the results in the Best Arm Identification problem from Section 5.2. Comparing vSSM+KF and vSSM+KF-u, there is a slight improvement in performance from using an input signal in the KF layer, but it is not statistically significant.

**vSSM+KF learns consistently across environments.** From the detailed results in Figure 14, we observe that vSSM+KF consistently improves performance over the Memoryless lower-bound and achieves the best or

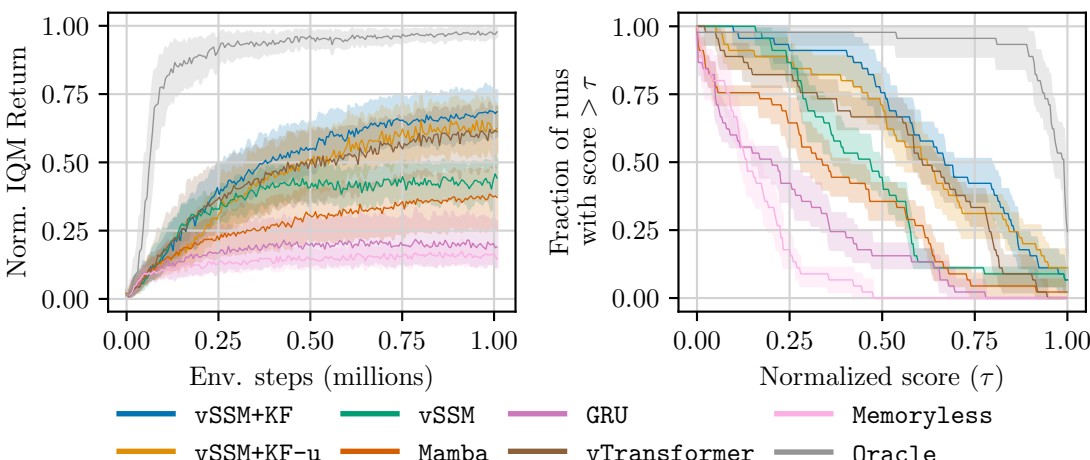

Figure 7: Aggregated performance in noisy DMC benchmark (9 tasks) with 95% bootstrap confidence intervals over five random seeds. **(Left)** Inter-quartile mean returns normalized by the score of `Oracle`. **(Right)** Performance profile after 1M environment steps. Higher curves correspond to better performance.

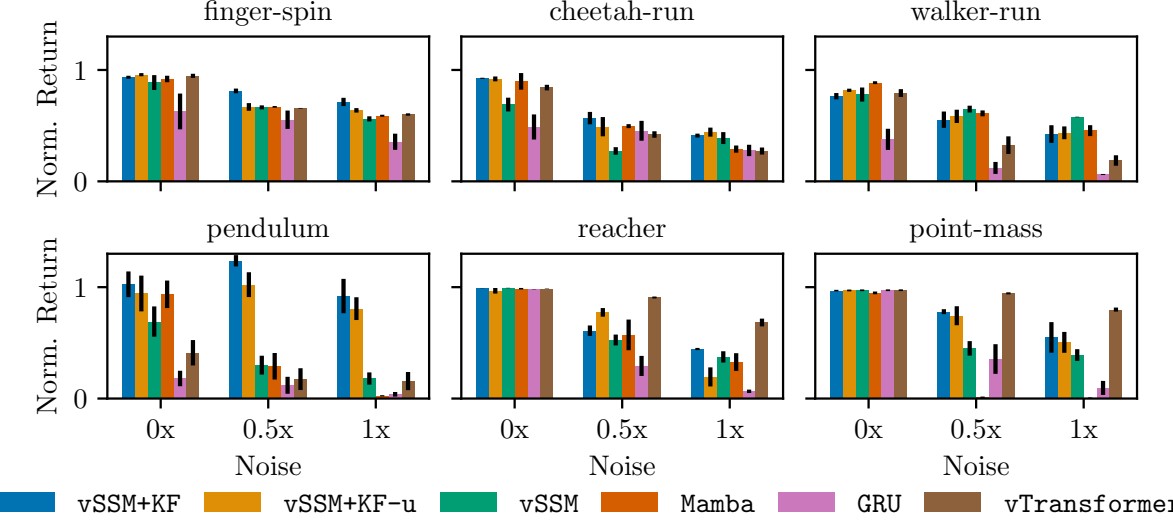

Figure 8: Final performance comparison of recurrent models in six tasks over increasing noise levels. We report the mean and standard error over five random seeds (ten for `pendulum` due to large variance) of the return after 1M environment steps, normalized by the score of `Oracle`.

comparable final performance in five out of nine tasks. Instead, `GRU`, `Mamba` and `vTransformer` completely fail to learn in some tasks, barely matching the performance of `Memoryless`.

We conduct an additional ablation over increasing noise levels in six representative tasks from the DMC suite, as shown in Figure 8. Training curves are included in Appendix G

**`vSSM+KF` performs close to `Oracle` under full observability.** We observe `vSSM+KF` generally matches the performance of `Oracle` in the absence of noise (normalized score close to 1.0), whereas `vSSM` and `vTransformer` significantly underperform in some tasks. This suggests that the added probabilistic filtering in `vSSM+KF` is a general-purpose strategy even under full observability.

**`vSSM+KF`'s robustness to noise is environment dependent.** Figure 8 suggests that robustness to noise depends generally on the environment, without any clear patterns related to task specifics. `vSSM+KF` is more

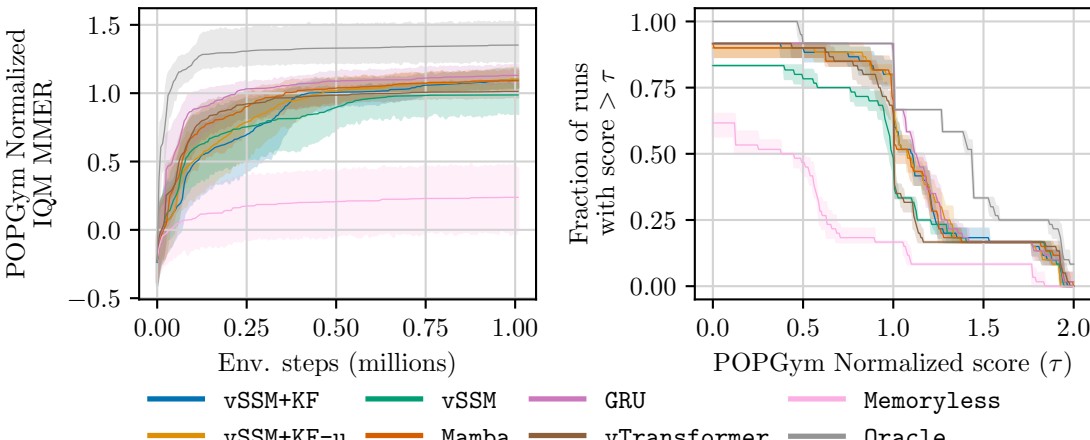

Figure 9: Aggregated performance in POPGym selected environments (12 tasks) with 95% bootstrap confidence intervals over five random seeds. We normalize the maximum-mean episodic return (MMER) by the best reported MMER in (Morad et al., 2023)**(Left)** Normalized IQM MMER **(Right)** Performance profile after 1M environment steps. Higher curves correspond to better performance and a score of 1.0 means equivalent performance as the best baseline (per environment) reported in POPGym.

robust in `finger-spin`, `cheetah-run` and `pendulum`[4], `vSSM` is more robust in `walker-run` but significantly underperforms in other environments, and `vTransformer` is more robuts in `reacher` and `point-mass` but fails to learn in `pendulum`. Overall, `vSSM+KF` shows the most consistent performance across environments and noise levels.

We include additional experiments with noisy DMC tasks in Appendix H, where we compare performance of `vSSM` and `vSSM+KF` against state-of-the-art model-based approaches. The main insight from this comparison is that our model-free approach mostly matches the performance of model-based methods *without* additional representation learning objectives.

### 5.4 General Memory Capabilities

So far the evaluations were conducted in tasks where probabilistic filtering was intuitively expected to excel. In this experiment, we evaluate performance in a wider variety of POMDPs from the POPGym (Morad et al., 2023) benchmark. We select a subset of 12 tasks that probe models for long-term memory, compression, recall, control under noise and reasoning. The aggregated results are shown in Figure 9 and full training curves are also included in Appendix I. Below we discuss the main insights.

**KF layers can be generally helpful in POMDPs.** From the performance profile in Figure 9 we observe a statistically significant gap between `vSSM` and `vSSM+KF`. Interestingly, the largest improvements in sample-efficiency (`RepeatPreviousEasy`) and final performance (`MineSweeperEasy`) correspond to tasks that probe for memory duration and recall, respectively. The parameter count difference between `vSSM` and `vSSM+KF` in these problems is less than 6%, so we believe model capacity is unlikely the reason behind the large performance difference. We hypothesize that, while probabilistic filtering is not required to solve these tasks, the KF layer has extra flexibility via the latent observation and noise signals to accelerate the learning process. We also highlight that `vSSM+KF` and `vSSM+KF-u` show comparable performance in this benchmark, suggesting the input signal to be less critical in general memory tasks.

**`vSSM+KF` is less sample-efficient in pure-memory tasks.** In particular, we observe that `Mamba`'s input-selectivity is the best-suited mechanism for SSM agents to solve long-term memory problems, matching the

---

[4]We found that `Oracle` underperforms in the noiseless `pendulum-swingup`, similarly reported in (Luis et al., 2023), which is why the normalized score in this task is larger than 1.0 in some cases. Moreover, performance does not strictly decrease under higher noise levels, perhaps because noise may actually help avoid early convergence under sparse rewards.

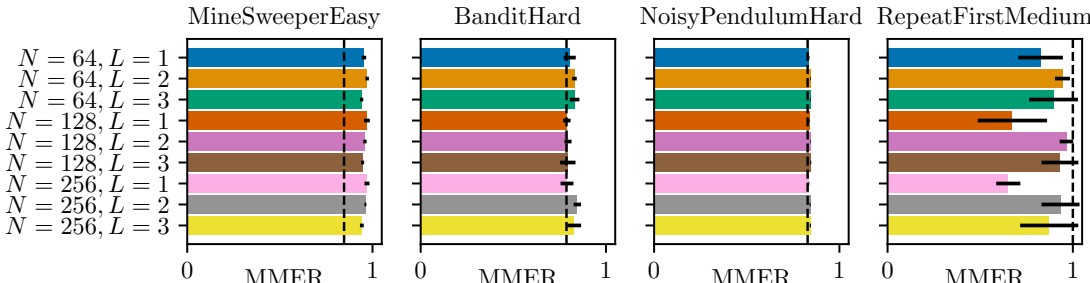

Figure 10: POPGym ablation for `vSSM+KF` over the latent state size $N$ and the number of layers $L$. We report the mean and standard error over five random seeds of the MMER score after 1M environment steps. The MMER score is shifted from $[-1, 1]$ to $[0, 1]$ for easier visualization. The vertical line represents the best score reported by Morad et al. (2023).

performace of `GRU` and `vTransformer`. This is an expected result based on the associative recall performance of Mamba reported in its original paper (Gu and Dao, 2023).

**Linear SSMs can have strong performance.** Morad et al. (2023) report poor performance when combining PPO with the S4D (Gu et al., 2022b) model. While we do not evaluate the S4D model and use an off-policy algorithm in our RAC architecture, our evaluation shows various linear SSMs have strong performance, often surpassing the best reported scores in Morad et al. (2023). Our observation is consistent with the strong performance of PPO with the S5 model reported by Lu et al. (2023).

### 5.5 Ablation

We conduct an ablation on `vSSM+KF` where we vary two hyperparameters: the latent state size $N$ and the number of stacked KF layers $L$[5]. We select four representative tasks from POPGym that test different memory capabilities. The final scores are presented in Figure 10 and the full training curves are included in Appendix K. Performance is most sensitive to these hyperparameters in the `RepeatFirstMedium` task, where the agent must recall information from the first observation over several steps. The general trend is that using more than one layer improves final performance and increases sample-efficiency (see the training curves in Figure 19). Our results are aligned with the good performance of stacked S5 layers reported by Lu et al. (2023), but differ from the observations in (Ni et al., 2023), where both LSTM and transformer models performed best with a single layer in a similar long-term memory task (T-maze passive). From these observations, we believe an interesting avenue for future work is to study what mechanisms enable effective stacking and combination of multiple recurrent layers.

## 6 Conclusion

We investigated the use of Kalman filter (KF) layers as sequence models in a recurrent actor-critic architecture. These layers perform closed-form Gaussian inference in latent space and output a *filtered* state representation for downstream RL components, such as value functions and policies. Thanks to the associative nature of the Kalman filter equations, the KF layers process sequential data efficiently via parallel scans, whose runtime scales logarithmically with the sequence length. To handle trajectories with variable length in off-policy RL, we introduced Masked Associative Operators (MAOs), a general-purpose method that augments any associative operator to recover the correct hidden state when processing padded input data. The KF layers are used as a drop-in replacement for RNNs and SSMs in recurrent architectures, and thus can be trained similarly in an end-to-end, model-free fashion for return maximization.

We evaluated and analysed the strengths and weaknesses of several sequence models in a wide range of POMDPs. KF layers excel in tasks where uncertainty reasoning is key for decision-making, such as the Best Arm Identification task and control under observation noise, significantly improving performance over

---

[5]We use an RMSNorm output block in `vSSM+KF` since it was critical to ensure stable learning when $L > 1$.

stateful models like RNNs and deterministic SSMs. In more general tasks, including long-term memory and associative recall, KF layers typically match the performance of transformers and other stateful sequence models, albeit with a lower sample-efficiency.

**Limitations and Future Work.** We highlight notable limitations of our methodology and suggest avenues for future work. First, we investigated two design decisions in KF layers related to time-varying process noise and posterior covariance as output features. While they resulted in worse performance (see Appendix E), in principle they generalize KF layers and may bring benefits in other tasks or contexts, so we believe it is worth further investigation. Second, we use models with relatively low parameter count ($< 1M$) which is standard in RL but not on other supervised learning tasks. It may be possible that deeper models with larger parameter counts enable new capabilities, e.g., probabilistic reasoning, without explicit probabilistic filtering mechanisms. Third, `vSSM+KF` uses KF layers as standalone history encoders, but more complex architectures may be needed to stabilize training at larger parameter counts. Typical strategies found in models like `Mamba` include residual connections, layer normalization, convolutions and non-linearities. Fourth, our evaluations were limited to POMDPS with relatively low-dimensional observation and action spaces, where small models have enough capacity for learning. Future work could further evaluate performance in more complex POMDPs (e.g., with image observations) and compare with our findings.

### Acknowledgments

We would like to thank Philipp Becker for providing an efficient parallel scan routine in Pytorch which we use on all our SSM-based experiments.

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

## A  Associativity of Masked Associative Operators

Let $\tilde{a}, \tilde{b}, \tilde{c} \in \tilde{\mathcal{E}}$ refer to elements in the space of the MAO $\tilde{\bullet}$, as in Definition 1, with $\tilde{a} = (a, m_a)$, $\tilde{b} = (b, m_b)$, $\tilde{c} = (c, m_c)$. We show that if the sequence $\{m_a, m_b, m_c\}$, is a right-padding mask, that is: $m_a = 1 \implies m_b = m_c = 1$, and $m_b = 1 \implies m_c = 1$, then it holds that $(\tilde{a} \,\tilde{\bullet}\, \tilde{b}) \,\tilde{\bullet}\, \tilde{c} = \tilde{a} \,\tilde{\bullet}\, (\tilde{b} \,\tilde{\bullet}\, \tilde{c})$, i.e., the MAO is associative. Similar to the proof in Lu et al. (2023) we consider all possible values for $\{m_a, m_b, m_c\}$.

**Case 1:** $m_b = 1$ **and** $m_c = 1$**.** The binary masks of $b$ and $c$ are on, so $\tilde{b} \,\tilde{\bullet}\, \tilde{c} = \tilde{b}$, $\tilde{a} \,\tilde{\bullet}\, \tilde{b} = \tilde{a}$ and $\tilde{a} \,\tilde{\bullet}\, \tilde{c} = \tilde{a}$. Then,

$$(\tilde{a} \,\tilde{\bullet}\, \tilde{b}) \,\tilde{\bullet}\, \tilde{c} = \tilde{a} \tag{8}$$
$$= \tilde{a} \,\tilde{\bullet}\, (\tilde{b} \,\tilde{\bullet}\, \tilde{c}) \tag{9}$$

**Case 2:** $m_b = 0$ **and** $m_c = 1$**.** The binary mask of $b$ if off while that of $c$ is on, so $\tilde{b} \,\tilde{\bullet}\, \tilde{c} = \tilde{b}$, then:

$$(\tilde{a} \,\tilde{\bullet}\, \tilde{b}) \,\tilde{\bullet}\, \tilde{c} = \tilde{a} \,\tilde{\bullet}\, \tilde{b} \tag{10}$$
$$= \tilde{a} \,\tilde{\bullet}\, (\tilde{b} \,\tilde{\bullet}\, \tilde{c}) \tag{11}$$

**Case 3:** $m_b = 0$ **and** $m_c = 0$**.** No mask is applied, then the MAO is equivalent to the underlying operator $\bullet$, which is associative by Definition 1.

Note the case $m_b = 1$ and $m_c = 0$ violates associativity, but it is impossible under our initial assumption of a right-padding mask sequence $\{m_a, m_b, m_c\}$.

## B  Implementation Details

In this section we provide details of various components of the RAC architecture and the specific implementations of history encoders. All methods are implemented in a common codebase written in the Pytorch framework (Paszke et al., 2019).

**Embedder.** We embed the concatenated observation-action history with a simple linear layer mapping from the combined observation-action dimension to the embedding dimension $E$.

**Soft Actor-Critic.** We use a standard SAC implementation with optional automatic entropy tuning (Haarnoja et al., 2019). For discrete action spaces, we use the discrete version of SAC by (Christodoulou, 2019) and one-hot encode the actions.

**vSSM, vSSM+KF & vSSM+KF-u.** These methods share a similar implementation, with an input linear layer, a linear recurrence and an output linear layer. vSSM is equivalent to only using the "Predict" block from the KF layer, while vSSM+KF-u removes the input signal $u_{:t}$. For all methods, we discretize the SSM using the zero-order hold method and a learnable scalar step size $\Delta$. In practice we use an auxiliary learnable parameter $\tilde{\Delta}$ and define $\Delta = \texttt{softplus}(\tilde{\Delta})$ to ensure a positive step size. as similarly done in Mamba. We initialize $\tilde{\Delta}$ with a negative value such that after passing through the softplus and after ZOH discretization, the SSM is initialized with eigenvalues close to 1 (i.e., slow decay of state information over time).

**Mamba.** Standard Mamba model from Gu and Dao (2023). We use a reference open-source implementation[6] and modify the parallel scan to use the associated MAO.

**GRU.** Standard implementation included in Pytorch.

**vTransformer.** Default implementation of a causal transformer encoder from Pytorch. We additionally include a sinusoidal positional encoding, as done in prior work using transformers for RL (Ni et al., 2023).

---

[6]https://github.com/johnma2006/mamba-minimal/tree/03de542a36d873f6e6c4057ad687278cc6ae944d

## C Hyperparameters

Table 1: Hyperparameters used for Section 5. For the `Mamba` parameters, we use the notation from the code by Gu and Dao (2023) and select parameters to match a effective state size $N = 128$. `GRU` and `vTransformer` use default parameters from Pytorch unless noted otherwise.

| Parameter | BestArm | DMC | POPGym |
|---|---|---|---|
| **Training** | | | |
| Buffer size | | $\infty$ | |
| Adam learning rate | | 3e-4 | |
| Env. steps | 500K | | 1M |
| Batch size | 64 | | 32 |
| Update-to-data (UTD) ratio | 0.25 | | 1.0 |
| # Eval episodes | 100 | | 16 |
| **RAC** | | | |
| Embedding size (E) | | 16 | |
| Latent size (N) | | 128 | |
| Activations | | ReLU | |
| Context length | 256 | | 64 |
| Actor MLP | [128] | | [256, 256] |
| Critic MLP | [256] | | [256, 256] |
| **SAC** | | | |
| Discount factor $\gamma$ | | 0.99 | |
| Entropy temp. $\alpha$ | 0.1 | | Auto |
| Target entropy (continuous) | N/A | | -dim($\mathcal{A}$) |
| Target entropy (discrete) | N/A | | $-0.7\log\big(1/\dim(\mathcal{A})\big)$[7] |
| **History Encoders (common)** | | | |
| Latent size $N$ | | 128 | |
| # layers | | 1 | |
| **vSSM, vSSM+KF & vSSM+KF-u** | | | |
| $\tilde{\Delta}$ init | | -7 | |
| **A** init | | HiPPO (diagonal) | |
| **B** init | | **I** | |
| $\mathbf{\Sigma}^{\mathrm{p}}$ init | | **I** | |
| Inital state belief | | $\mathcal{N}(\mathbf{0}, \mathbf{I})$ | |
| RMSNorm output? | No | Yes | No |
| **Mamba** | | | |
| **A** init | | HiPPO (diagonal) | |
| `d_model` (embedding size) | | 16 | |
| `d_state` (per-channel hidden size) | | 4 | |
| Expand factor $E$ | | 2 | |
| Size of $\Delta$ projection | | 1 | |
| 1D Conv kernel size | | 4 | |
| **vTransformer** | | | |
| # heads | | 1 | |
| Feedforward size | 128 | | 256 |

---

[7]We use a lower value of $-0.35\log\big(1/\dim(\mathcal{A})\big)$ in the `MineSweeper` environment from POPGym, as the default value resulted in divergence during training.

# D    Best Arm Identification Training Curves

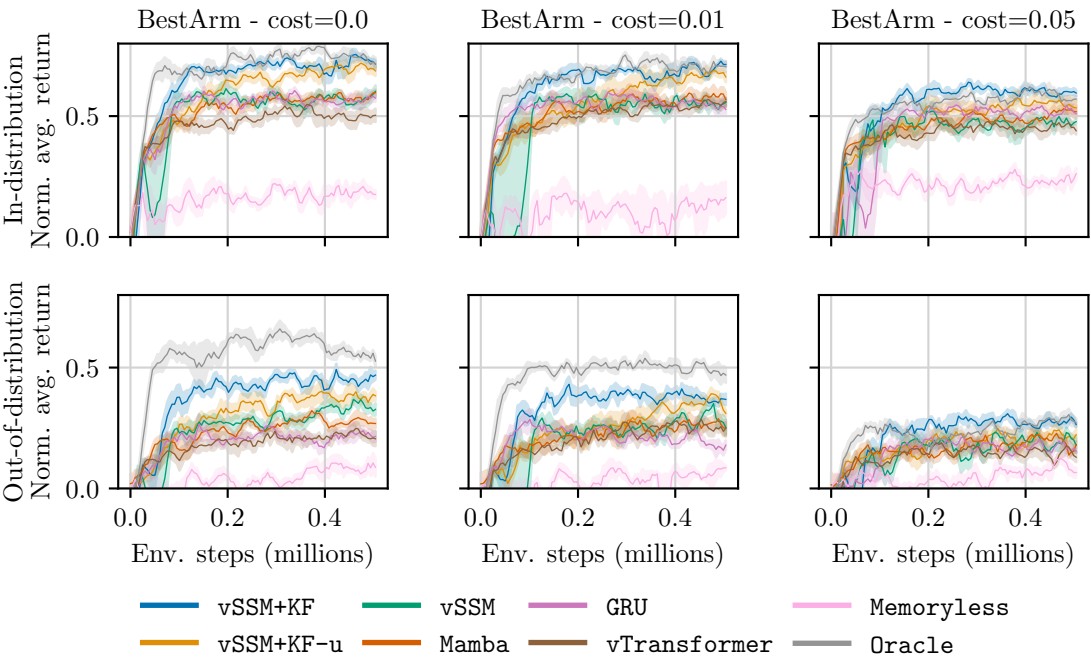

Figure 11: Normalized average return over 100 episodes in and out of distribution, for increasing costs. We report the mean and standard error over 5 random seeds.

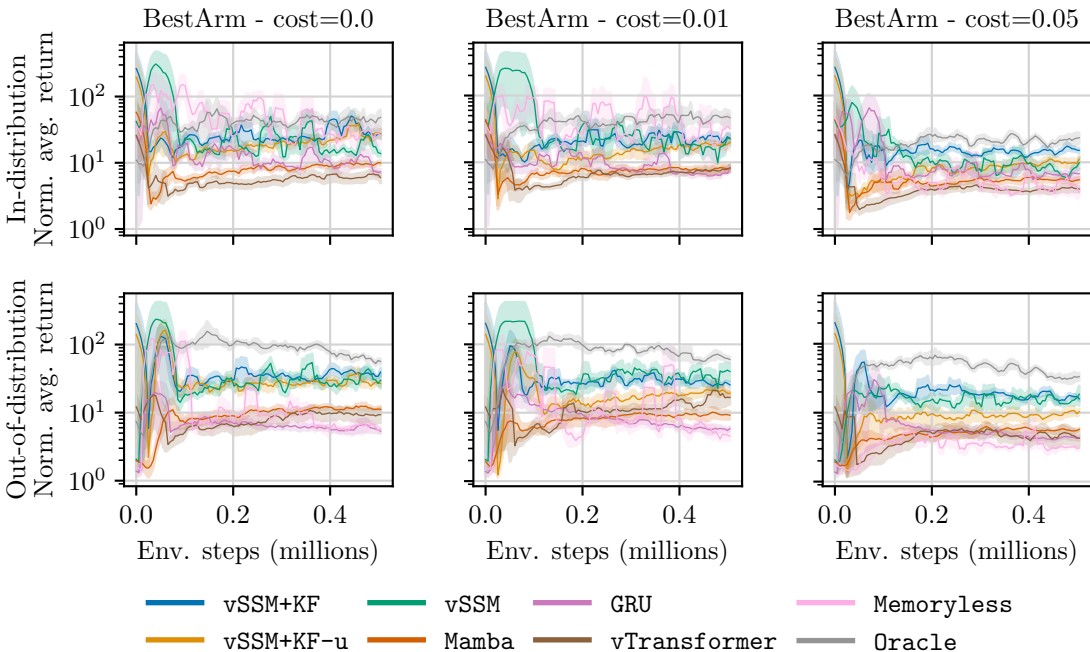

Figure 12: Average (log) episode length over 100 episodes in and out of distribution, for increasing costs. We report the mean and standard error over 5 random seeds.

# E  KF Layer Design Ablation

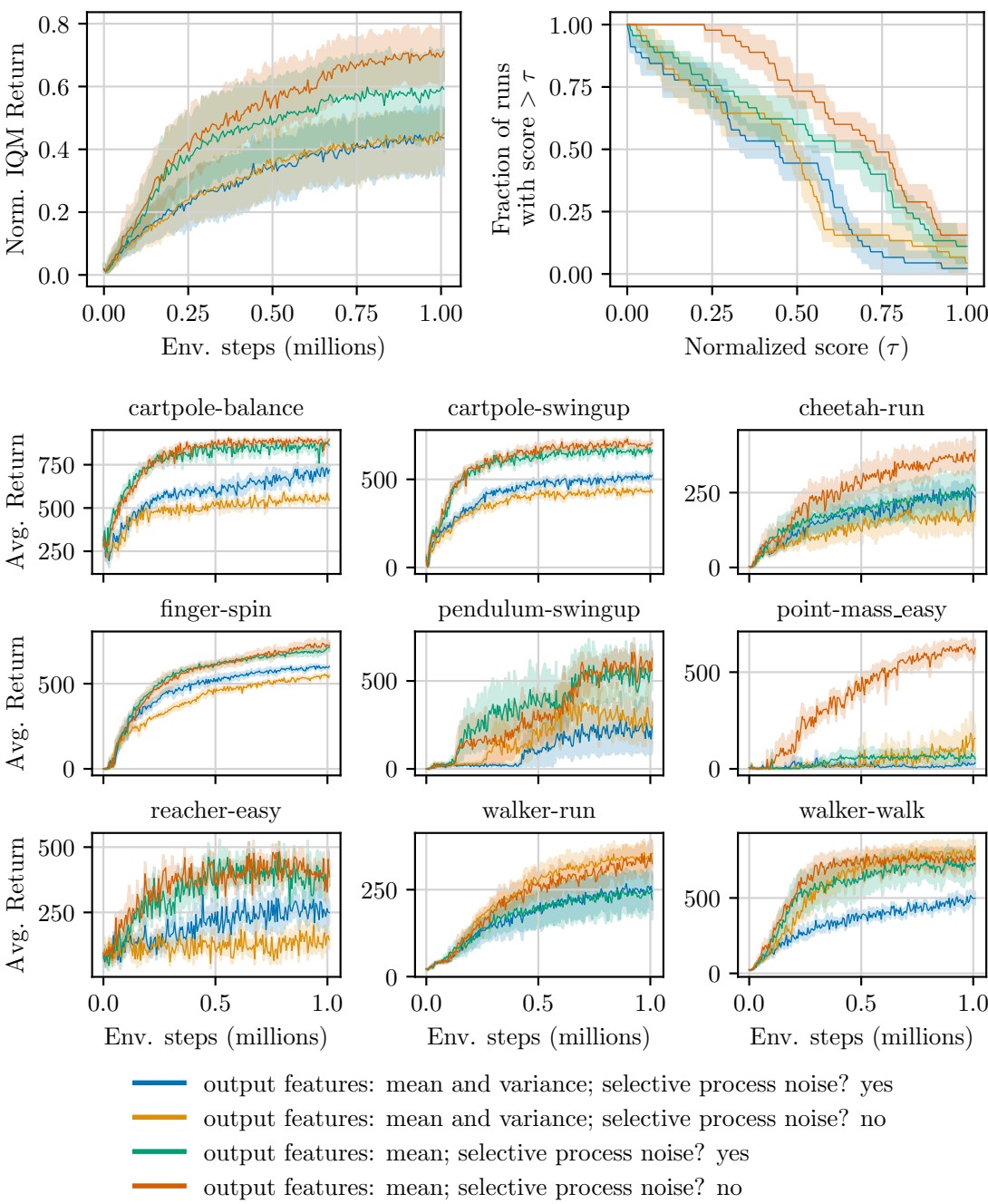

Figure 13: Ablation on design considerations for KF layers. **(Top)** Aggregated performance in noisy DMC benchmark (9 tasks) with 95% bootstrap confidence intervals over five random seeds. **(Top-Left)** Inter-quartile mean returns normalized by the score of `Oracle`. **(Top-Right)** Performance profile after 1M environment steps. **(Bottom)** Training curves. We show mean and standard error over five random seeds. Based on these results, our final design for the KF layer uses only the posterior mean state as the output feature and a time-invariant process noise.

## F  DMC Training Curves

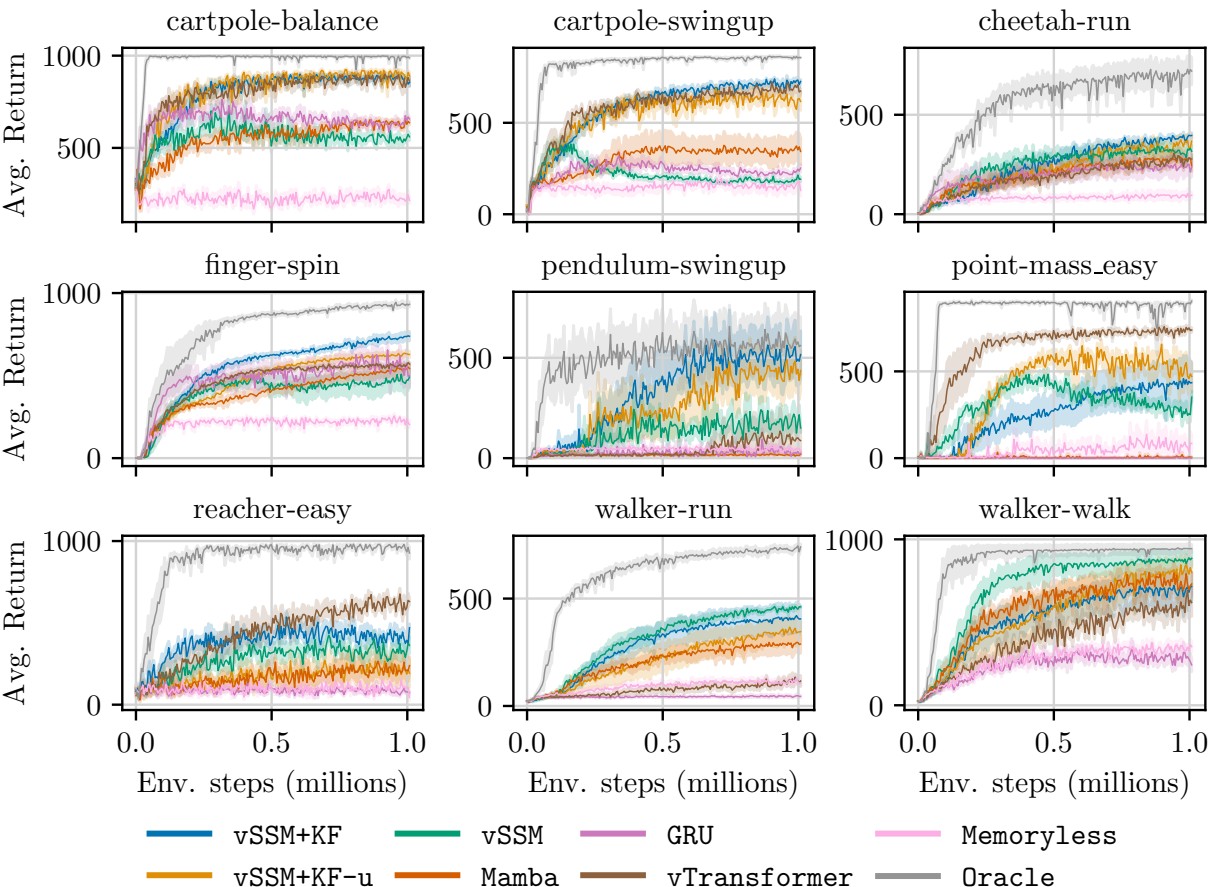

Figure 14: Training curves for the noisy DMC benchmark. We show mean and standard error over five random seeds. For all tasks, we add zero-mean Gaussian noise to the observations with a scale of 0.3, except the `pendulum-swingup` and `point-mass` where the scale is 0.1.

## G  DMC Noise Ablation

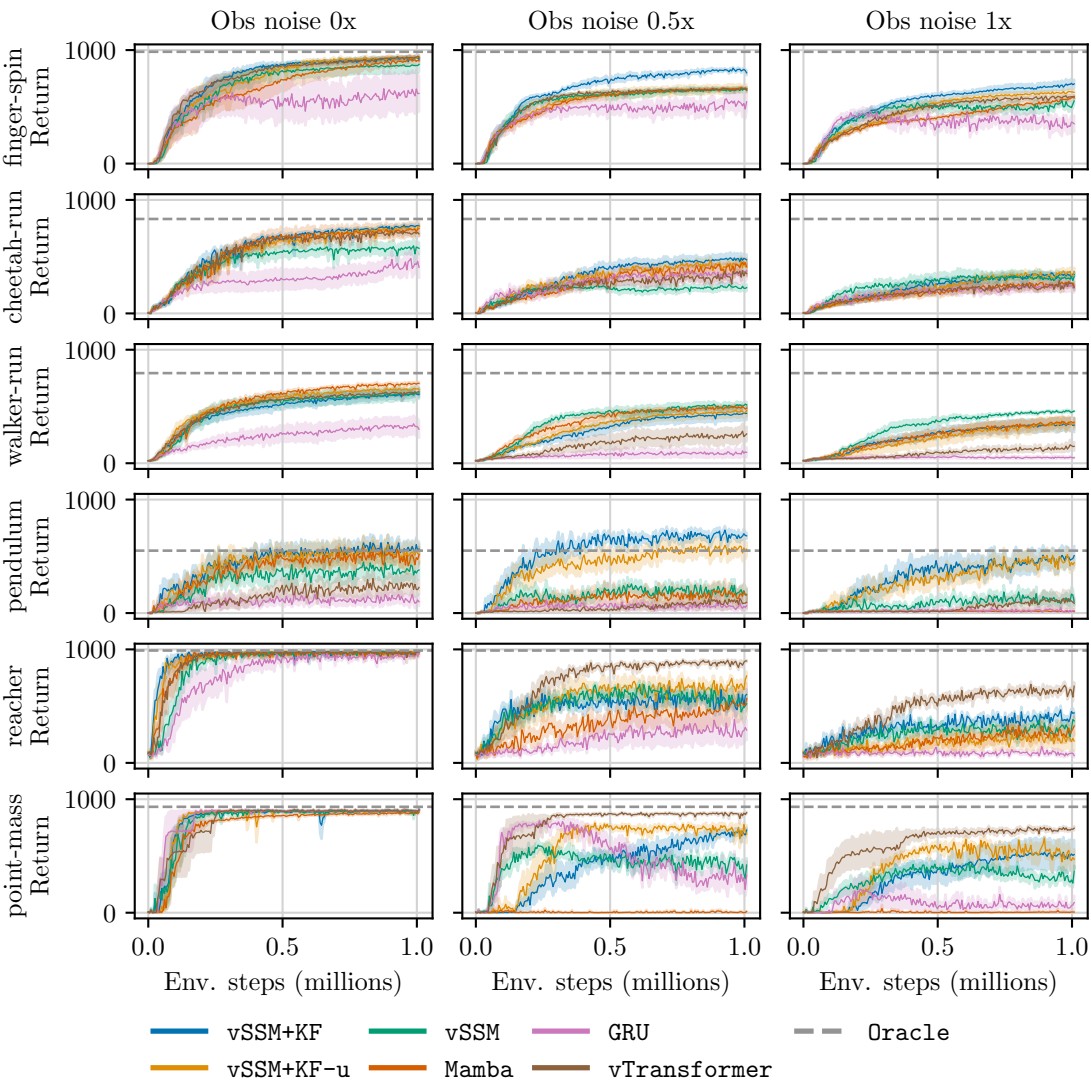

Figure 15: Training curves in six environments from the DMC benchmark with increasing levels of noise. We show mean and standard error over five random seeds (ten for `pendulum`). The base noise scale for all tasks is 0.3, except the `pendulum-swingup` and `point-mass` environments where the scale is 0.1

## H  DMC Comparison to Model-Based Approaches

In Figure 16, we compare performance between `vSSM`, `vSSM+KF` and the following model-based baselines reported in Becker et al. (2024)[8]:

**Kalmamba.** Uses a Mamba (Gu and Dao, 2023) backbone that outputs the SSM matrices, which are then used within the VRKN architecture proposed in Becker and Neumann (2022). The representation is trained end-to-end in a variational loss to produce plausible dynamic predictions. The learned representation is then frozen and used within a SAC policy optimizer.

---

[8]The experimental data was provided by the authors on personal communication.

**RSSM+SAC.** The Recurrent SSM as proposed in Hafner et al. (2019). Similarly, the representation is trained on a variational loss for reconstruction of observations, while the policy is trained with SAC using the learned latent representation. The gradients from SAC are not used to update the representation.

**VRKN+SAC** The Variational Recurrent Kalman Network (Becker and Neumann, 2022). Similarly to RSSM+SAC and Kalmamba, the representation is trained with a variational loss for reconstruction while the policy uses the learned representation for control.

**SAC.** Equivalent to the `Memoryless` baseline; a SAC agent with no memory.

The benchmark includes both observation *and* action noise with a standard deviation of $\sigma = 0.3$.

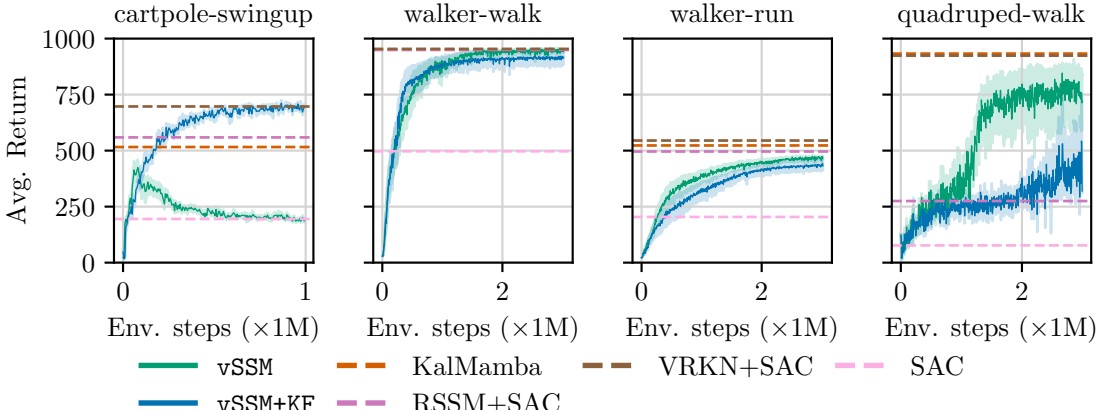

Figure 16: Comparison of performance in DMC environments with observation and action noise. For the model-based baselines, we report the final performance after 1M environment steps, as reported in Becker et al. (2024).

From Figure 16 we observe that in three out of four tasks, `vSSM+KF` is close to or matches the asymptotic performance of the best model-based baseline, albeit with less sample-efficiency. These results highlight that good performance can be achieved in these tasks without the representation learning objectives from model-based approaches. In `quadruped-walk` we found that probabilistic filtering hurts performance, given the performance difference between `vSSM+KF` and `vSSM`. We hypothesize that to handle the larger observation space in `quadruped-walk` ($\sim 4\times$ larger than `walker-run`), `vSSM+KF` would require further hyperparameter tuning and potential regularization techniques we do not explore in this work. Moreover, the sample-efficiency of the architecture could be further improved using recent developments such as higher update-to-data ratios and network resets (D'Oro et al., 2022).

# I  POPGym Training Curves

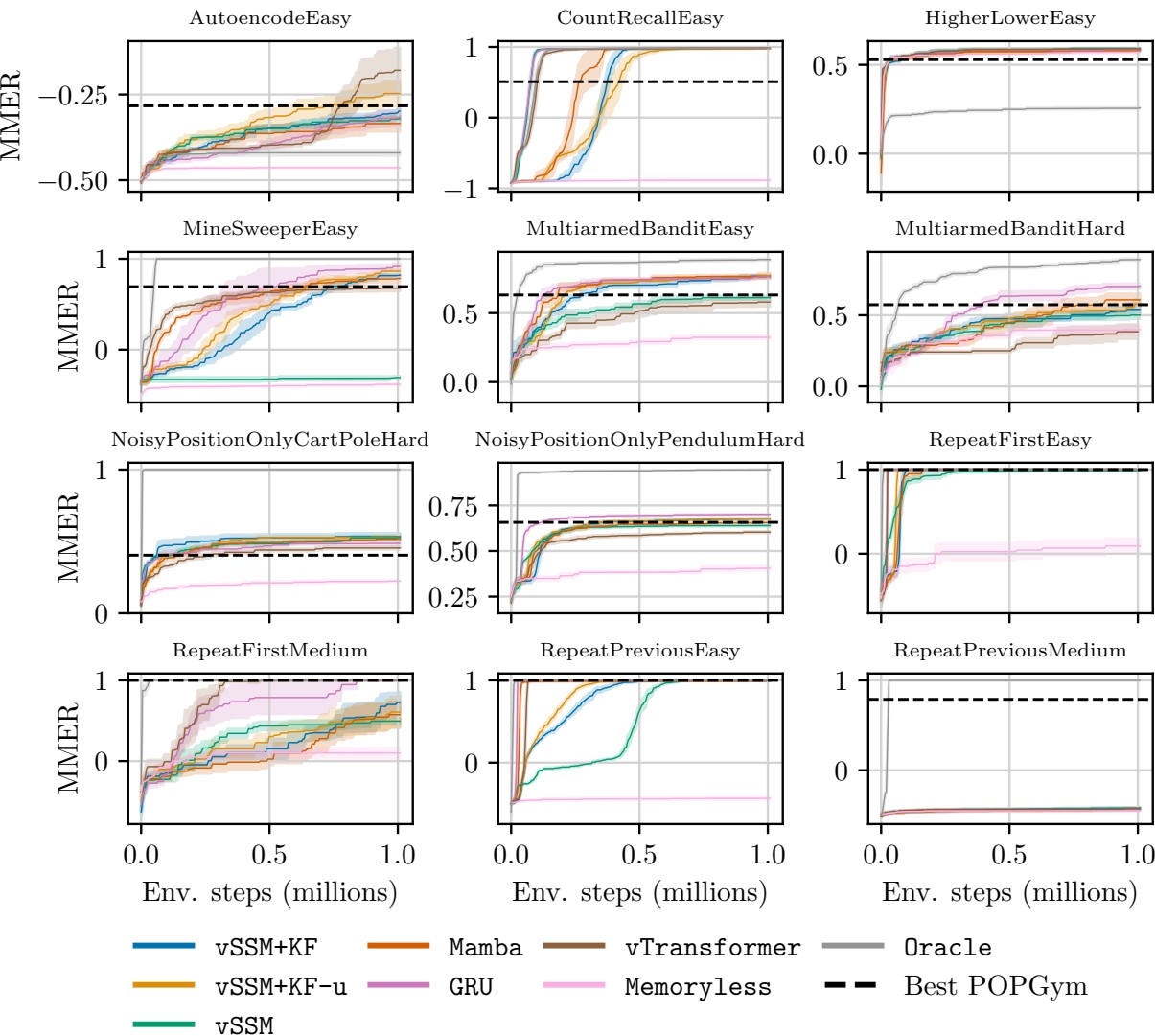

Figure 17: POPGym training curves. We show mean and standard error over five random seeds.

# J POPGym Scores

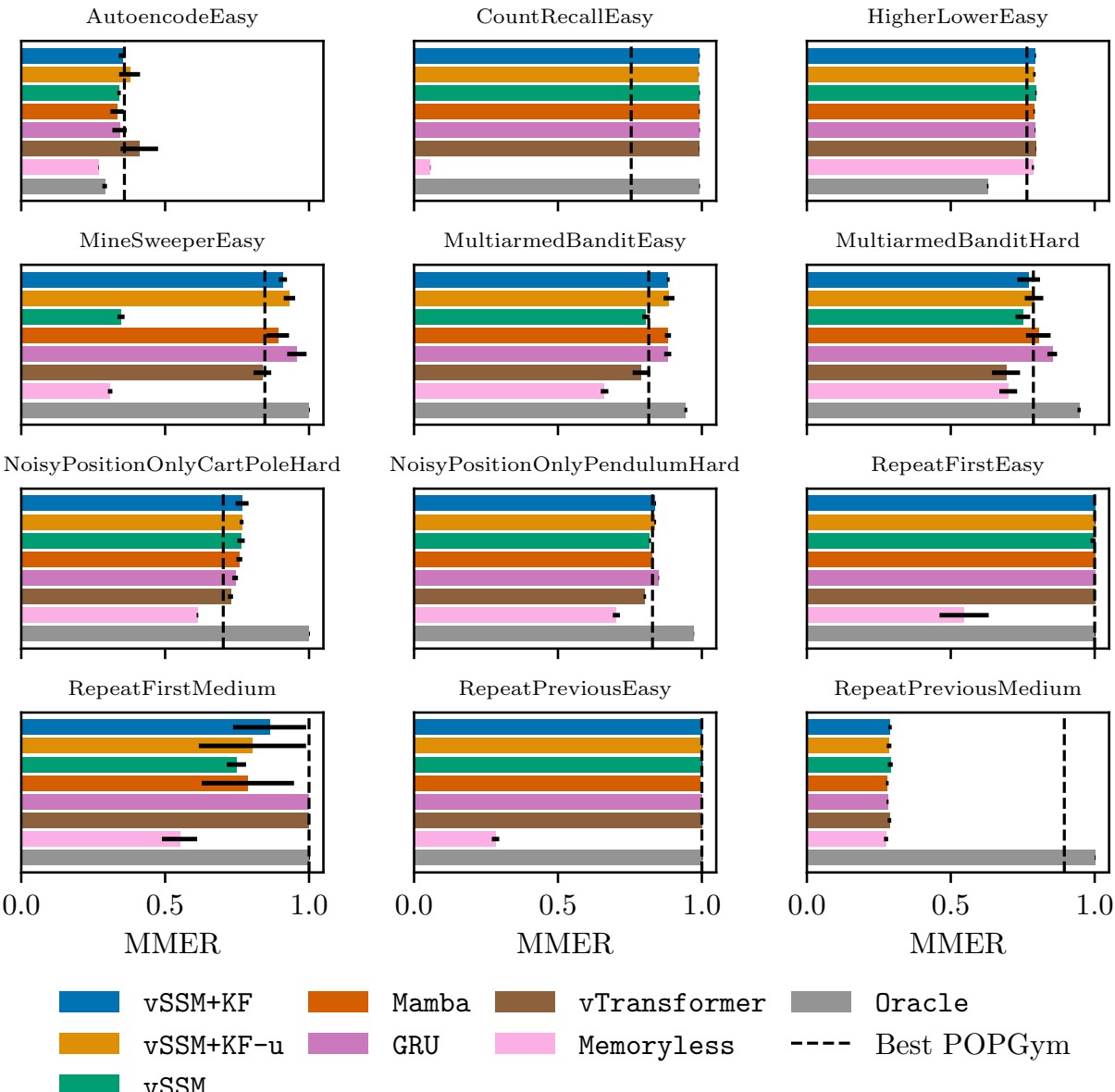

Figure 18: POPGym final MMER after 1M training steps. We show mean and standard error over five random seeds.

Table 2: Scores on POPGym tasks after 1M environment steps. For each environment, we report the MMER mean and standard error over 5 random seeds after 1M steps of training. The MMER is calculated from 16 test episodes. For reference, we include the best MMER score reported by Morad et al. (2023) (mean and standard deviation over three random seeds). We **bold** the highest score(s) per environment obtained by a sequence model.

| | AutoencodeEasy | CountRecallEasy | HigherLowerEasy | MineSweeperEasy |
|---|---|---|---|---|
| vSSM+KF | $-0.299 \pm 0.012$ | $\mathbf{0.983} \pm 0.002$ | $0.588 \pm 0.002$ | $0.818 \pm 0.014$ |
| vSSM+KF-u | $-0.247 \pm 0.036$ | $\mathbf{0.978} \pm 0.001$ | $0.581 \pm 0.004$ | $0.864 \pm 0.020$ |
| vSSM | $-0.320 \pm 0.006$ | $\mathbf{0.984} \pm 0.002$ | $\mathbf{0.592} \pm 0.003$ | $-0.307 \pm 0.012$ |
| Mamba | $-0.335 \pm 0.023$ | $0.982 \pm 0.002$ | $0.580 \pm 0.003$ | $0.783 \pm 0.039$ |
| GRU | $-0.317 \pm 0.025$ | $\mathbf{0.984} \pm 0.001$ | $0.586 \pm 0.002$ | $\mathbf{0.916} \pm 0.034$ |
| vTransformer | $\mathbf{-0.179} \pm 0.065$ | $0.982 \pm 0.001$ | $\mathbf{0.590} \pm 0.001$ | $0.676 \pm 0.031$ |
| Oracle | $-0.420 \pm 0.008$ | $0.984 \pm 0.002$ | $0.257 \pm 0.003$ | $1.000 \pm 0.000$ |
| Memoryless | $-0.463 \pm 0.001$ | $-0.887 \pm 0.001$ | $0.571 \pm 0.004$ | $-0.382 \pm 0.008$ |
| Best POPGym | $-0.283 \pm 0.029$ | $0.509 \pm 0.062$ | $0.529 \pm 0.002$ | $0.693 \pm 0.009$ |

| | BanditEasy | BanditHard | NoisyCartPoleHard | NoisyPendulumHard |
|---|---|---|---|---|
| vSSM+KF | $\mathbf{0.766} \pm 0.005$ | $0.541 \pm 0.040$ | $\mathbf{0.535} \pm 0.023$ | $0.677 \pm 0.003$ |
| vSSM+KF-u | $\mathbf{0.771} \pm 0.019$ | $0.579 \pm 0.032$ | $\mathbf{0.531} \pm 0.007$ | $0.675 \pm 0.003$ |
| vSSM | $0.612 \pm 0.013$ | $0.501 \pm 0.025$ | $0.528 \pm 0.013$ | $0.639 \pm 0.004$ |
| Mamba | $\mathbf{0.764} \pm 0.011$ | $0.608 \pm 0.043$ | $\mathbf{0.516} \pm 0.010$ | $0.658 \pm 0.009$ |
| GRU | $\mathbf{0.763} \pm 0.012$ | $\mathbf{0.705} \pm 0.017$ | $0.486 \pm 0.010$ | $\mathbf{0.701} \pm 0.001$ |
| vTransformer | $0.580 \pm 0.030$ | $0.384 \pm 0.049$ | $0.454 \pm 0.009$ | $0.604 \pm 0.004$ |
| Oracle | $0.889 \pm 0.005$ | $0.892 \pm 0.006$ | $1.000 \pm 0.000$ | $0.946 \pm 0.001$ |
| Memoryless | $0.324 \pm 0.013$ | $0.399 \pm 0.031$ | $0.225 \pm 0.003$ | $0.406 \pm 0.012$ |
| Best POPGym | $0.631 \pm 0.014$ | $0.574 \pm 0.049$ | $0.404 \pm 0.005$ | $0.657 \pm 0.002$ |

| | RepeatFirstEasy | RepeatFirstMedium | RepeatPreviousEasy | RepeatPreviousMedium |
|---|---|---|---|---|
| vSSM+KF | $\mathbf{1.000} \pm 0.000$ | $0.726 \pm 0.127$ | $\mathbf{1.000} \pm 0.000$ | $\mathbf{-0.423} \pm 0.006$ |
| vSSM+KF-u | $\mathbf{1.000} \pm 0.000$ | $0.607 \pm 0.186$ | $\mathbf{1.000} \pm 0.000$ | $\mathbf{-0.429} \pm 0.008$ |
| vSSM | $0.989 \pm 0.009$ | $0.495 \pm 0.034$ | $\mathbf{1.000} \pm 0.000$ | $\mathbf{-0.420} \pm 0.008$ |
| Mamba | $\mathbf{1.000} \pm 0.000$ | $0.575 \pm 0.160$ | $0.993 \pm 0.001$ | $-0.441 \pm 0.005$ |
| GRU | $\mathbf{1.000} \pm 0.000$ | $\mathbf{1.000} \pm 0.000$ | $\mathbf{1.000} \pm 0.000$ | $-0.440 \pm 0.004$ |
| vTransformer | $\mathbf{1.000} \pm 0.000$ | $\mathbf{1.000} \pm 0.000$ | $\mathbf{1.000} \pm 0.000$ | $\mathbf{-0.426} \pm 0.006$ |
| Oracle | $1.000 \pm 0.000$ | $1.000 \pm 0.000$ | $1.000 \pm 0.000$ | $1.000 \pm 0.000$ |
| Memoryless | $0.093 \pm 0.085$ | $0.100 \pm 0.061$ | $-0.434 \pm 0.013$ | $-0.450 \pm 0.007$ |
| Best POPGym | $1.000 \pm 0.000$ | $1.000 \pm 0.000$ | $1.000 \pm 0.000$ | $0.789 \pm 0.288$ |

## K   POPGym Ablation

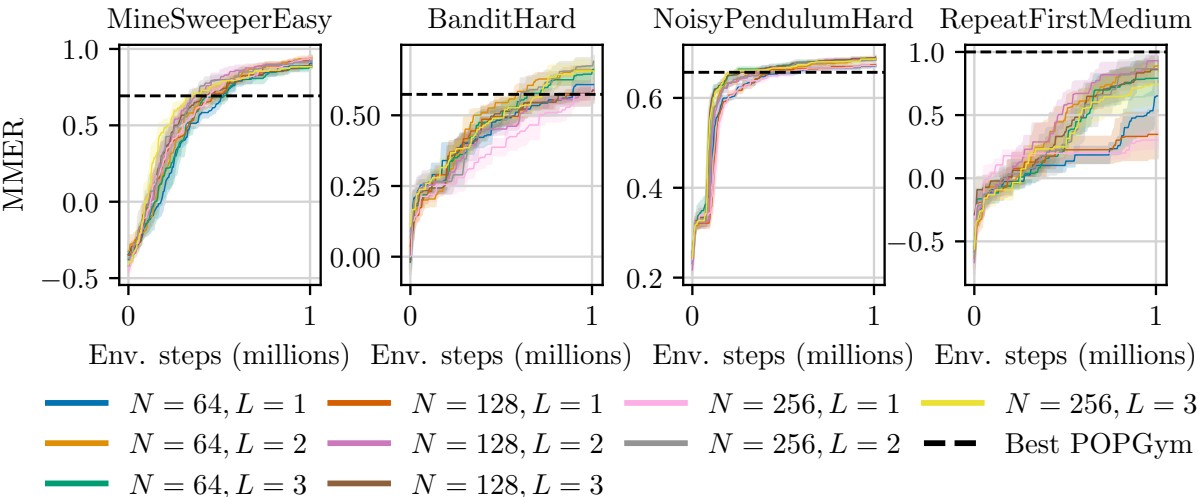

Figure 19: POPGym training curves for `vSSM+KF` ablation over latent state size $N$ and number of KF layers $L$. We show mean and standard error over five random seeds. For this experiment, `vSSM+KF` uses an RMSNorm output block to ensure stability for $L > 1$.

