# OpenReview forum: "Uncertainty Representations in State-Space Layers for Deep Reinforcement Learning under Partial Observability"
_TMLR — Accepted by TMLR_

### Review · Reviewer_AeJf · 2024-10-26

**Summary Of Contributions:**

The paper shows that augmenting a model-free algorithm with a Kalman filter layer allows it to deal with partial observability thanks to improved uncertainty reasoning capabilities.

**Audience:**

Yes

**Broader Impact Concerns:**

No ethical concerns.

**Claims And Evidence:**

Yes

**Requested Changes:**

1. A comparison with a different sota methodology would allow the reader to better understand the potential of the proposed method.

**Strengths And Weaknesses:**

# Strengths
1. The paper is easy to follow thanks to a clear structure and careful explanations.
2. Using a kalman filter in a POMDP setting is novel and has good empirical performances.
3. The methodoogy is thoroughly evaluated in the paper.

# Weaknesses
1. The main tool to asses the proposed methodology is using ablations.

---

> ### Author Response · Authors · 2024-11-19
>
> We thank the reviewer for their constructive feedback and for acknowledging key strengths of our paper.
>
> > A comparison with a different sota methodology would allow the reader to better understand the potential of the proposed method.
>
> We appreciate the suggestion from the reviewer. We would like to understand if there was a specific methodology in mind. In the paper we follow standard empirical practices found in state-of-the-art methods for RL under POMDPs. These include: (1) testing with five random seeds in various environments (a total of 22 environments across three distinct benchmarks) (2) using robust statistical metrics to gauge performance like the inter-quartile mean, bootstrapped confidence intervals and performance profiles, (3) including typical baselines like oracles, memoryless agents and a wide range of history encoders, including stateful (GRU, Mamba, SSMs) and stateless (transformer) models.
>
> Beyond these established practices, we conduct carefully designed experiments to understand capabilities of our method. In the Best Arm Identification environment, we deviate from the commonly found methodologies and provide in-depth analysis using post-training evaluations, which allow for a thorough analysis of the agent’s adaptation patterns in comparison to the Bayesian oracle. Moreover, we also include discussions and insightful visualizations on a concrete adversarial episode in this environment where KF layers avoid pitfalls found in SSMs without probabilistic filtering.
>
> If any details about our methodology are confusing or not evident, we would appreciate it if the reviewer could point them out. We are happy to clarify any missing details in a revision of our paper.

---

> > ### Comment · Reviewer_AeJf · 2024-11-24
> >
> > Thanks for the clarification on the evaluation practices in POMPDs RL. I now understand better the scope and how the paper is related with the existing literature.
> >
> > I suggest to accept the paper as it is (including the important changes suggested by the other reviewers).

---

### Review · Reviewer_CzkF · 2024-11-03

**Summary Of Contributions:**

The paper introduces a history encoder utilizing Kalman filtering layers for simple probabilistic inference on the latent state within a model-free RL architecture. This work is reported to be the first empirical evaluation of such probabilistic inference layers. The authors explore the integration of model-free RL techniques with structured state space models (SSMs), combining deterministic and probabilistic SSMs with inference techniques. They investigate the performance implications of these layers across various benchmark settings, thereby expanding the traditional applications of RL and Kalman filters.

**Audience:**

Yes

**Claims And Evidence:**

No

**Requested Changes:**

As part of the review process, I have carefully evaluated the manuscript and appreciate the efforts put forth by the authors. However, to enhance the quality and impact of the paper, I believe it requires a major revision that addresses several critical concerns outlined in my comments. Addressing these issues will not only strengthen the manuscript but also align it more closely with the standards expected in high-caliber publications.

For ease of reference and to ensure a comprehensive revision, I have summarized the main issues that need attention. I expect that the concerns raised will be thoughtfully addressed, and the revisions will incorporate substantial changes in the following areas:

Requested Changes:
1.	Improve Paper Organization:
Reorganize to clearly present the paper's objectives and main contributions in appropriate sections, such as the introduction or conclusion.

2.	Enhance Literature Review:
Broaden the scope of the related work section to include significant contributions from disciplines like signal processing and control, not just machine learning.

3.	Reduce Bias Towards Previous Work:
Integrate a wider range of studies and reduce the heavy reliance on Becker’s work to ensure a balanced and comprehensive approach.

4.	Clarify Theoretical Modeling:
Clearly differentiate between foundational model assumptions and simplifying assumptions for tractability, and discuss the implications of these choices.

5.	Specify Algorithm Justifications and Comparisons:
Justify the choice of specific algorithms used and compare them against a broader range of state-of-the-art methods, including theoretical guarantees.

6.	Validate Experimental Design:
Expand the scope of experiments to include alternative strategies and provide benchmarks against state-of-the-art methods to ensure robust evaluation.

7.	Address Minor Issues:
Review and address minor issues as noted in the feedback, such as inconsistencies in writing style and terminology

**Strengths And Weaknesses:**

Strengths:
1.	The paper integrates a Kalman filtering-type algorithm with model-free reinforcement learning, potentially opening new avenues in RL applications.
2.	It contributes new insights into the application of these methods within the context of model-free RL.
3.	The use of the Deep Mind Control (DMC) suite supports an extensive experimental setup that provides a broad evaluation across multiple scenarios, showcasing the adaptability and potential of the proposed methods.


Major Weaknesses:

Paper Organization and Placement of Information: The organization of the manuscript requires improvement. Essential elements such as the paper's objectives and main contributions are not positioned effectively for clear comprehension. For instance, critical statements like, "In this paper, we propose a history encoder implemented via Kalman filtering layers that perform simple probabilistic inference on the latent state," are unexpectedly located in the background section. Such a pivotal statement should ideally be highlighted either at the end of the introduction or as a conclusive remark in the theoretical background to guide the reader smoothly into the core of the paper.

Related Work and Literature Review:
The related work section of the manuscript is notably brief, spanning only about 1.3 pages in a 28-page document. While it primarily focuses on literature from the machine learning community, featuring contributions from well-known conferences like NeurIPS and ICML, it significantly overlooks the rich and longstanding contributions from other disciplines that have extensively engaged with the Kalman Filter, such as signal processing, control, and information fusion.
The discussion revisits earlier studies on Deterministic SSMs, Probabilistic SSMs, and Inference in Linear SSMs but falls short of critically evaluating the underlying assumptions of these works or providing a thorough perspective on these topics.
This section could be greatly enhanced by broadening its scope to include a wider range of literature, especially emphasizing diverse applications and developments related to DNN implementations of the Kalman Filter. Such an expansion would not only offer a more comprehensive and informative background but also better reflect the interdisciplinary impact and breadth of current research in this field. Moreover, a more inclusive literature review would set the stage for a more solid and informative empirical benchmark that effectively compares with the state of the art.

Bias Towards Prior Work:
The manuscript demonstrates a considerable bias towards the work of Becker, repeatedly referencing it to the exclusion of other significant contributions in the field. This focus suggests a narrow design choice and a lack of familiarity with a broader range of relevant literature. While building on previous research is standard practice, the extent of reliance on Becker’s work in a Transactions paper of this magnitude raises concerns. Such reliance should be balanced with a broader integration of diverse studies to ensure a comprehensive approach that enriches the paper’s contributions and relevance.

Self-Contained Scope:
For a contribution to be valuable in a high-caliber publication, it must be both comprehensive and self-contained. The current manuscript's approach limits its scope and diminishes its potential to serve as a definitive reference on the topic. Although I am familiar with Becker's work, this narrow focus might alienate readers who are not well-versed in these particular studies, consequently narrowing the paper’s appeal.

To enhance its accessibility and relevance, the manuscript should strive for a more autonomous exposition that robustly supports its findings and arguments without excessive reliance on prior work—unless such work is indeed foundational and universally recognized within the field.

Theoretical Modeling Clarity and Evaluation of Novelty:
To robustly evaluate whether a solution is truly novel or state-of-the-art, a clear understanding of all model assumptions is essential, including any simplifying assumptions made for tractability. These can significantly narrow the scope of problems for which the algorithm is effective. It is crucial that the manuscript distinctly outlines these assumptions, allowing for a precise assessment of the algorithm’s applicability and limitations.

Furthermore, to confirm if an algorithm represents a genuine advancement in the field, it must be benchmarked against not only minor variations of the same type but also against the known state-of-the-art for the test case or relevant theoretical guarantees. This broader comparison is vital to accurately gauge the true impact and innovation of the proposed solution.

Isolating the problem formulation from the solution design choices will help to correctly identify the relevant benchmarks and theoretical bounds for performance and clarify the true significance of the paper and its contribution to the broader community. Additionally, a true decoupling of necessary model assumptions from design choices will help mitigate bias in the authors' approach, potentially leading to more innovative and effective design choices.

The manuscript currently fails to adequately separate modeling or model assumptions from algorithms and inference techniques, which undermines its scholarly rigor. A robust problem formulation, distinct from the solution technique, is essential in the background section to ensure a comprehensive and unbiased exploration of solutions.

The paper would significantly benefit from a clear initial presentation of problem formulation, explicitly stating the optimization objectives and modeling assumptions—particularly detailing how the physical world is modeled and how agents interact within it. The model statement should remain independent of the class of algorithms used.

By adhering to these guidelines, the manuscript can enhance its scholarly rigor and provide a clearer, more impactful contribution to the field. This structured approach will ensure the research is comprehensively evaluated for both its methodological soundness and its innovative value.

Example from the paper:
The manuscript does not effectively differentiate between foundational model assumptions, which describe the world or class of problems being modeled, and simplifying assumptions made for tractability. Understanding the implications of these choices is crucial. Additionally, such simplifying assumptions, including the use of structured SSMs and diagonal matrices discussed in subsection 3.4’s Kalman Filter analysis, should be explicitly noted in the problem formulation section. These choices potentially narrow the class of problems that can be addressed, and if these assumptions do not impact the class of problems or expected performance, it should be clearly justified.

Evaluation of Novelty and Design Choices:
The paper asserts a novel implementation of a history encoder via Kalman filtering layers for model-free RL, claiming, "To the best of our knowledge, this is the first work that empirically evaluates the performance of such probabilistic inference layers in a model-free RL architecture." While the application in model-free RL is appreciated, the novelty of this approach raises several critical questions:

1. Optimality of Design Choice: The paper leverages a specific type of Kalman Filter algorithm, heavily based on Becker’s work without apparent modification or improvement. It is essential to discuss why this particular algorithm was chosen over other learned Kalman Filter algorithms that might also fulfill these roles. The omission of a broad range of pertinent literature in this context raises concerns about the comprehensiveness of the empirical evaluation.

2. Trade-offs in Simplifying Assumptions: What are the performance trade-offs involved in restricting the Kalman Filter to a linear type? Additionally, what complexities are reduced by these assumptions, and how do these choices impact the overall effectiveness of the solution?

3. Theoretical and Empirical Validation: Is the assumption that a Kalman Filter type algorithm would be an optimal history encoder under all circumstances valid? The manuscript would benefit from more robust theoretical arguments and empirical evidence supporting this premise.

4. State-of-the-Art Comparison: Does the final algorithm as implemented stand up as state-of-the-art for the test cases provided in the empirical sections, or are there gaps in performance when compared to existing solutions?

While focusing on a specific algorithm from the outset and exploring its effectiveness is acceptable, such intent should be explicitly stated. This approach sets realistic expectations about the scope and novelty of the research but may limit perceived innovation unless significant enhancements or insights are demonstrated.


In-Depth Analysis Suggestion:
The proposed history encoder relies on a Kalman Filter, which is known to be optimal for linear Gaussian models. To better understand how your implementation compares to classical solutions, such as in LQG control scenarios where the Kalman Filter is the optimal tracker and LQR is the optimal controller, testing your methodology in these settings could be insightful.

Given the high-level architecture agreement that a Kalman Filter is suitable and acknowledging that a linear Kalman Filter is a reasonable choice, exploring how alternative learnable Kalman Filters could integrate into or enhance the proposed design would be valuable. A comparative analysis of these implementations would help clarify whether the design choices made, including simplifying assumptions for better computational efficiency, do not unduly compromise the solution's effectiveness.
These discussions and analyses are crucial for validating the design choices made and for ensuring that the proposed solution not only advances the field by integrating novel approaches but also stands rigorous evaluation against established methodologies.

Experiments Evaluation:
I commend the effort and resources dedicated to conducting the experiments presented in this paper, as they significantly contribute to its richness. However, I hold several reservations about the experimental design and its implications:

1. Relevance of RL Approaches: The paper predominantly focuses on reinforcement learning (RL) based strategies. However, not all benchmarks necessitate a full RL approach, and this narrow focus may omit crucial opportunities to benchmark against more diverse and possibly superior state-of-the-art methodologies. It would be beneficial if the paper expanded its scope to include broader problem formulations. This would provide a more balanced view, contrasting RL solutions against the best available solutions for the specific problems addressed or exploring test cases where assumptions integral to other methods are challenged.


2. Evaluation Scope and Comparisons: The statement, “we evaluate the RAC architecture under different history encoders,” suggests a limited exploration confined to variations within a predetermined design framework, specifically focused on a particular type of history encoders. For each test case examined, it is crucial to clarify the state of the art and how it compares to your architecture, irrespective of the high-level design. Where applicable, I am interested in seeing a performance gap analysis relative to an oracle, which could provide insights into the maximum potential improvement over current results.

3. Benchmarking Integrity: Concerning the benchmark algorithms used for comparative analysis, there is an underlying assumption that these are optimally trained and tuned. It is imperative to confirm whether these benchmarks are indeed reflective of the best possible configurations as evidenced by previously published results or through the use of pre-trained models. This validation is essential to ensure that the comparisons made are fair and that the conclusions drawn about the efficacy of the proposed architecture are sound.

By addressing these points, the paper can enhance its contribution to the field, offering a more comprehensive evaluation that not only tests the proposed methods against current best practices but also provides a clear benchmark for future innovations. This approach will not only solidify the validity of the experimental results but also potentially elevate the perceived novelty and applicability of the research.

Regarding the MAB Experiment 5.2:
The significant gap to the oracle in the Multi-Armed Bandit (MAB) experiment raises questions about the chosen methods and their effectiveness. Given that bandit problems are classically addressed with state-of-the-art solutions that do not necessitate the use of neural networks or extensive reinforcement learning, but rather simpler bandit policies, this discrepancy merits closer examination.
I suggest conducting an additional experiment where the setting involves a noisy observed state that aligns more closely with typical bandit scenarios, avoiding the need for an embedding from a textual user recommendation. This approach would allow for a more controlled assessment of performance and enable a direct comparison with state-of-the-art solutions tailored to this specific problem type. Such an experiment would not only test the robustness of the proposed methods under more typical conditions but also provide a clearer benchmark against established techniques.

Furthermore, it would be beneficial to include a discussion on where similar experiments have been conducted, outlining the results obtained in those studies and how they compare to the findings presented here. Understanding these comparisons is crucial for contextualizing the results and assessing the true novelty and impact of the experimental approach used in this study.
By addressing these points, the paper could greatly enhance its empirical credibility and provide a more solid foundation for the claims of innovation and effectiveness in tackling MAB problems.

Regarding the Control Experiments 5.3:
Control theory is a thoroughly researched area, yet the manuscript lacks references to pertinent control literature. For a more comprehensive evaluation, it would be beneficial to know where similar experiments have been conducted and how those results compare to the findings presented here. When possible, comparing these results to the optimal benchmarks is essential to ascertain the effectiveness of the proposed methods.

The manuscript mentions the use of the Deep Mind Control (DMC) suite but fails to provide detailed context about this reference. For the paper to be as self-contained and accessible as possible, more information is needed. Specifically, it should clarify whether the observations involve noisy visual frames or noisy state variables. Additionally, an understanding of the underlying physical model used in these experiments would greatly aid in appreciating the constraints and limitations of the modeling approach employed.

Minor Weaknesses:

1.	Writing Consistency: The paper exhibits inconsistencies in the presentation of acronyms, with some capitalized and others not, such as "reinforcement learning (RL)," "state-space models (SSMs)," "Kalman filter (KF) layer," and "Markov Decision Process (MDP)." To improve readability and maintain a professional tone, I suggest standardizing the format in which acronyms are introduced and utilized throughout the document.

2.	Mathematical Rigor: The manuscript describes dynamical systems using differential equations, which lacks rigor in the context of the extensive literature that employs stochastic differential equations for such descriptions. Considering there is no further reference to continuous-time systems, it may be more appropriate to omit this less rigorous description and commence directly with a discussion on discrete-time systems.

3.	Overly Specific Methodological Focus: Subsection 3.1, titled "RL in POMDP," seems narrowly focused on specific reinforcement learning methods. This could limit the paper's appeal and applicability, suggesting a need for a broader examination of relevant techniques within this context.

4.	Technical Clarifications: Regarding the statement “vSSM+KF. Probabilistic SSM via the KF layers described in Figure 2, which is equivalent to adding Kalman filtering on top of vSSM,” it would be more accurate to specify that this integration involves adding an update step, which clarifies the actual technical process involved.

5.	Consistency in Terminology: The paper uses the terms "running example" and "motivating example" interchangeably. For clarity and to enhance the paper's structure, sticking to one term and consistently using it throughout would be beneficial. Additionally, labeling this example clearly would aid in precise references throughout the discussion.

---

> ### Author Response · Authors · 2024-11-19
> **Response to Reviewer CzkF (1/3)**
>
> We thank the reviewer for the comprehensive and constructive feedback on our manuscript. Below we answer key questions and comment on the main concerns.
>
> ## Strengths
> We thank the reviewer for highlighting important strengths of our paper.
>
> > The use of the Deep Mind Control (DMC) suite supports an extensive experimental setup that provides a broad evaluation across multiple scenarios, showcasing the adaptability and potential of the proposed methods.
>
> We agree that our DMC experiments evaluate the capabilities of the different history encoders to learn good representations for control from noisy observations. However, we also test more broadly both in established benchmarks (POPGym), as well as newly proposed problems (Best Arm). In total, the paper includes over 1000 RL training runs, across three benchmarks and 8 different baselines. We believe the breadth and thoroughness of our experiments support our claims and shine some light on our research questions. In particular, the Best Arm Identification problem is a simple, yet effective example of a task where the inductive bias encoded through the KF layer is instrumental to achieve good performance.
>
> ## Improve Paper Organization
> We thank the reviewer for their suggestion and have revised the paper based on it.
>
> > Essential elements such as the paper's objectives and main contributions are not positioned effectively for clear comprehension.
>
> We have revised our introduction to clearly present our objectives and contributions.
>
> > critical statements like, "In this paper, we propose a history encoder implemented via Kalman filtering layers that perform simple probabilistic inference on the latent state," are unexpectedly located in the background section.
>
> We have removed this sentence from the background and instead convey the same information in the contribution statement of the Introduction.
>
> ## Enhance Literature Review
> We agree with the feedback from the reviewer and have substantially expanded our literature review and discussion. The revised version includes a broader scope of work around probabilistic state-space models and Kalman filters with neural networks, with discussions on their assumptions and how it relates to our work.
>
> ## Reduce Bias Towards Previous Work
> We understand the reviewer’s concern regarding bias towards Becker’s work. In the revised version of the paper we made great efforts to mitigate this bias by (1) extensively expanding the related work section and discussing a broader range of prior work, and (2) having a more comprehensive and self-contained background section.
>
> ## Self-Contained Scope
> We have revised our background section to be more self-contained. In particular, we included a subsection “Simplifying Assumptions” that clearly states the simplifications made on the linear-Gaussian SSM and, consequently, on the Kalman filter equations. The tradeoffs of such simplifications are discussed in the Related Work section.
>
> ## Clarify Theoretical Modeling
> We agree with the reviewer’s point about clearly separating modelling assumptions for tractability (like diagonal covariance matrices) from specific design choices (like the output features of the KF layer). First, we now include discussions on such simplifying assumptions as part of the related work on Kalman filters. Second, we added a “Simplifying assumptions” subsection in the Background to clearly explain the simplifications that were made and the resulting simplified KF equations.
>
> > Furthermore, to confirm if an algorithm represents a genuine advancement in the field, it must be benchmarked against not only minor variations of the same type but also against the known state-of-the-art for the test case or relevant theoretical guarantees.
>
> We understand the raised concern and have included comparisons against the recent Kalmamba method as well, as suggested by Reviewer FQjn. However, we want to highlight that the narrow scope of our empirical evaluation is a conscious decision: we tradeoff scope in favor of carefully controlled experiments, which ultimately enables us to rigorously validate the claims in the paper.
>
> > A robust problem formulation, distinct from the solution technique, is essential in the background section to ensure a comprehensive and unbiased exploration of solutions.
>
> We acknowledge the need for a precise problem formulation that is agnostic to the type of solutions proposed in the paper. We want to clarify that our paper focuses exclusively on the RL problem under partial observability. In Section 3.1, we introduce the formal problem statement of finding the optimal policy in a POMDP, which is standard in the RL literature (e.g., Ni et al. 2024). This problem statement is independent of the architecture we propose or the specific history encoders we evaluate.

---

> > ### Author Response · Authors · 2024-11-19
> > **Response to Reviewer CzkF (2/3)**
> >
> > ## Specify Algorithm Justifications and Comparisons
> >
> > > It is essential to discuss why this particular algorithm was chosen over other learned Kalman Filter algorithms that might also fulfill these roles
> >
> > We have included relevant discussions on this point in the Kalman filter paragraph of the Related Work section.
> >
> > > Trade-offs in Simplifying Assumptions: What are the performance trade-offs involved in restricting the Kalman Filter to a linear type? Additionally, what complexities are reduced by these assumptions, and how do these choices impact the overall effectiveness of the solution?
> >
> > These questions are now discussed both in the Related Work section and also when introducing the assumptions.
> >
> > > State-of-the-Art Comparison: Does the final algorithm as implemented stand up as state-of-the-art for the test cases provided in the empirical sections, or are there gaps in performance when compared to existing solutions?
> >
> > The POPGym results (a total of 12 different tasks) in Figure 9 show that our method performs, on average, on par with the best baseline methods from the POPGym paper [1], which includes a total of 13 sequence models in a similar model-free RL architecture. This is remarkable given the POPGym paper reports poor results using SSMs. In Figures 16-17 and Table 2 we show specific performance per-task of our POPGym benchmark.
> >
> > For the other benchmarks (Best Arm and DMC) we could not find relevant empirical results in the model-free literature, which is why we implemented several baselines to compare against our proposed approach, including transformers, RNNs and various SSMs.
> >
> > [1] POPGym: Benchmarking Partially Observable Reinforcement Learning, Morad et al. 2023.
> >
> > > While focusing on a specific algorithm from the outset and exploring its effectiveness is acceptable, such intent should be explicitly stated
> >
> > The reviewer raises an excellent point and we absolutely agree with it. Indeed, the intent of our paper is to focus on a specific algorithm and understand the role of probabilistic inference layers within it. We acknowledge this key point was not highlighted enough in the original version of the paper. The revised Introduction and Related Work section adds discussions and clarifications on this topic. We hope the revised paper clarifies the scope of our paper and why it remains highly relevant to the RL community.
> >
> > ## Validate Experimental Design
> >
> > > The paper predominantly focuses on reinforcement learning (RL) based strategies.
> >
> > We want to emphasize the paper sets out to answer very concrete questions around using probabilistic inference mechanisms in model-free RL to tackle partial observability. Therefore, from the onset, the paper narrows down its scope to reinforcement learning strategies exclusively.
> >
> > > The statement, “we evaluate the RAC architecture under different history encoders,” suggests a limited exploration confined to variations within a predetermined design framework, specifically focused on a particular type of history encoders
> >
> > We acknowledge that the paper explores a variety of history encoders within a fixed architecture. The paper is designed with such a scope to thoroughly investigate our research question regarding the use of probabilistic inference in model-free RL for POMDPs. However, we argue that our empirical evaluation considers a wide range of history encoders. All history encoders share the same inputs and outputs, a decision made to maintain consistency and fairness in our evaluations, but their internal mechanisms to process sequences are significantly different. Transformers use self-attention, GRUs use a gating mechanism within a non-linear recurrent model, Mamba uses input-based selectivity resulting in a time-varying SSM and vSSM+KF uses closed-form probabilistic inference in linear-Gaussian models. These methods cover a wide range of sequence-processing mechanisms reflective of state-of-the-art methods in RL, thus enriching our empirical evaluation and comparisons.
> >
> > > Where applicable, I am interested in seeing a performance gap analysis relative to an oracle, which could provide insights into the maximum potential improvement over current results.
> >
> > We want to emphasize that all our empirical results include information relative to an oracle. In Figures 4,5,7 and 9, we include the “Oracle” explicitly in the results. In Figure 8, the reported results are normalized by the score of the Oracle (so that scores below 1.0 signify a gap w.r.t the oracle).

---

> > > ### Author Response · Authors · 2024-11-19
> > > **Response to Reviewer CzkF (3/3)**
> > >
> > > > Benchmarking Integrity: Concerning the benchmark algorithms used for comparative analysis, there is an underlying assumption that these are optimally trained and tuned
> > >
> > > We understand the concern raised by the reviewer. It is indeed difficult to determine whether a specific algorithm is optimally trained and tuned. Since doing a large hyperparameter sweep for all the baselines and benchmarks we considered is computationally intractable, we adopted the following methodology to ensure a fair comparison: (1) all algorithms are implemented in the same codebase, share common architectural components and are trained under the same procedure, (2) most hyperparameters are shared across all history encoders, as shown in Appendix C. Moreover, several hyperparameters are chosen from prior work, such as [1]. (3) given our controlled setup, the amount of trainable parameters for all history encoders are within the same order of magnitude. Beyond the controlled experimental setup, we report results using robust statistics as suggested in [2].
> > >
> > > [1] Recurrent Model-Free RL Can Be a Strong Baseline for Many POMDPs, Ni et al. 2022
> > > [2] Deep Reinforcement Learning at the Edge of the Statistical Precipice, Agarwal et al. 2021
> > >
> > > Re: Section 5.2:
> > > >  I suggest conducting an additional experiment where the setting involves a noisy observed state that aligns more closely with typical bandit scenarios, avoiding the need for an embedding from a textual user recommendation.
> > >
> > > We believe the wording on this section caused misunderstandings and modified it in the revised version. The experiments we conducted were already as the reviewer suggests: the RL agent receives noisy (scalar) samples from the underlying bandit distribution and not some embedding from a textual user recommendation. Our motivating example about the recommendation system was only illustrative and not an actual description of the experiment we conducted. We have modified the relevant parts of the paper to avoid this confusion.
> > >
> > > ## Address Minor Issues
> > > We appreciate the detailed feedback and have modified the paper to incorporate all the suggestions.
> > >
> > > > Overly Specific Methodological Focus: Subsection 3.1, titled "RL in POMDP," seems narrowly focused on specific reinforcement learning methods. This could limit the paper's appeal and applicability, suggesting a need for a broader examination of relevant techniques within this context.
> > >
> > > We understand our methodological focus is narrow, but we argue it is an acceptable tradeoff to ensure a proper empirical evaluation of the paper research questions. Namely, we investigate whether probabilistic inference in state-space models brings benefits in model-free architectures. We believe the scope of the paper remains highly relevant to the RL community: (1) we use an architecture adopted broadly in the literature to tackle general POMDPs (including meta-RL), which means that our findings can have a large impact in these settings, (2) we evaluate commonly used sequence models (RNNs, transformers, SSMs), which highlights strengths and weaknesses of these models, and (3) our research questions and results open up exciting research avenues for leveraging probabilistic inference in model-free RL for POMDPs.
> > >
> > > Beyond the relevance of our setting, our claims are thoroughly supported and discussed with experiments. In particular, we took special care in controlling confounding factors in our empirical methodology: everything is implemented in the same codebase, sharing all architectural components except for the history encoders. Similarly, we use the same common hyperparameters and control the parameter count under different history encoders.

---

### Review · Reviewer_FQjn · 2024-11-10

**Summary Of Contributions:**

This paper discusses the limitations of current reinforcement learning architectures, such as recurrent neural networks, deterministic state-space models, and transformers, in handling environments with partial observability due to their lack of mechanisms to manage uncertainty in hidden state representations. To address this, the authors propose integrating a Kalman filter layer into model-free architectures. This Kalman filter layer conducts closed-form Gaussian inference in linear state-space models and can be trained end-to-end to maximize returns. Unlike traditional sequence models, the Kalman filter layer uses a parallel scan for processing, which enhances scalability by scaling logarithmically with sequence length. By incorporating explicit mechanisms for probabilistic filtering, the Kalman filter layers can directly replace other recurrent layers in typical model-free setups, offering significant advantages in scenarios where understanding uncertainty is crucial for decision-making. The effectiveness of this approach is demonstrated through experiments in various tasks with partial observability, where Kalman filter layers outperform other stateful models.

**Audience:**

No

**Broader Impact Concerns:**

No concerns.

**Claims And Evidence:**

No

**Requested Changes:**

- **Inclusion of Model-Based Comparisons:**
  Adding a discussion about why a model-free version is necessary alongside or instead of a model-based system like KalMamba would provide clarity. Including scenarios where the model-free approach provides distinct advantages would strengthen the paper's contribution.

- **Expanded Benchmarking:**
  Broadening the scope of benchmark comparisons to include a range of state-space models would provide a more comprehensive understanding of where the proposed method stands relative to existing techniques.

- **Enhanced Realism in Experimental Design:**
  Employing more complex and realistic task environments for testing, such as DMC tasks with image-based observations and variable backgrounds, would better assess the method’s practical efficacy and denoising capabilities.

**Strengths And Weaknesses:**

Your observations about the paper's strengths and weaknesses provide important insights that could significantly enhance the depth and impact of the research. Here’s a structured feedback that could be included in a review for the authors:

### Strengths

1. **Comprehensive Evaluation:**
   The paper's extensive experimental evaluations underscore its robustness, particularly highlighted in scenarios like the AI chatbot case. This extensive testing across different scenarios not only demonstrates the effectiveness of the proposed methods but also helps validate the approach under various conditions.

2. **Scalability and Efficiency:**
   The implementation of Kalman filter layers to process sequential data efficiently is a major strength. The ability to scale logarithmically with the sequence length could be pivotal for real-time applications and handling large datasets, which is crucial in modern AI systems.

### Weaknesses

1. **Comparison with KalMamba:**
   The paper introduces a model-free version of what seems to be conceptually similar to KalMamba, yet lacks a clear rationale or motivation for preferring a model-free approach over a model-based one. A comparative analysis between these two methodologies could clarify the benefits or trade-offs involved, particularly concerning performance impacts. It would be helpful to discuss situations where a model-free approach may outperform model-based systems and vice versa.

2. **Lack of Comprehensive Benchmarks:**
   The absence of direct comparisons with existing methods, both deterministic SSMs like CURL and probabilistic SSMs like DreamerV3, is a significant limitation. These comparisons are essential to position the new method within the landscape of existing research, providing a clearer understanding of its advantages or potential drawbacks.

3. **Simplistic Task Environments:**
   The noisy DMC tasks used in the experiments, characterized merely by true states plus Gaussian noise, may not adequately challenge the denoising capabilities of the proposed method. This setup might oversimplify the problem, making it easier for the model to perform well without truly demonstrating its effectiveness in more complex, real-world-like scenarios. It is suggested to utilize image-based observations with noisy backgrounds, which better mirror the challenges present in real-world applications, to test the robustness and utility of the proposed model under more demanding conditions.

---

> ### Author Response · Authors · 2024-11-19
> **Response to Reviewer FQjn (1/2)**
>
> We thank the reviewer for the constructive and detailed feedback on our work. Below we answer key questions and comment on the raised concerns.
>
> ## Strengths
> We thank the reviewer for highlighting some important strengths of our work.
>
> > The paper's extensive experimental evaluations underscore its robustness, particularly highlighted in scenarios like the AI chatbot case.
>
> We agree with the reviewer and would like to highlight the relevance of the Best Arm Identification task and our results. First, it provides a thorough overview of an agent’s capabilities to reason over uncertainty, adapt across episodes and generalize to unseen task parameters. Second, since it is a conceptually simple problem, it enables thorough interpretation of results, like the adaptation patterns and failure cases presented in the paper. Third, we observe some of the insights from this benchmark translate to other, more complex problems like the DMC suite with observations noise. For all of these reasons, we believe our results are highly relevant to the RL community and can serve as a starting point to develop and test uncertainty-aware RL agents.
>
> ## Comparisons to model-based approaches
> > The paper introduces a model-free version of what seems to be conceptually similar to KalMamba, yet lacks a clear rationale or motivation for preferring a model-free approach over a model-based one. A comparative analysis between these two methodologies could clarify the benefits or trade-offs involved, particularly concerning performance impacts.
>
> We understand the concern of the reviewer and would like to clarify the motivations behind our model-free formulation (read below). Moreover, we are currently running experiments to compare performance against the results reported in the Kalmamba paper, which include several model-based approaches. We would appreciate the reviewer’s patience and hope to post an updated version of the paper once they are finished.
>
> The discussion that follows has been summarized and included as part of the introduction and related work sections of the paper:
>
> The original intent of the paper is not to argue in favour or against model-free approaches compared to model-based ones. The motivation behind our paper is to understand the role of probabilistic inference methods in model-free architectures for POMDPs, since prior work in similar architectures only considered sequence models without such mechanisms (e.g., RNNs, transformers and deterministic SSMs). We found this curious, given the widespread use of probabilistic sequence models in the model-based approaches. Given this interesting gap in the literature, we thus framed our research questions to bring new understanding in this area. Our hypothesis, as stated in the introduction, is that such probabilistic tools may similarly benefit the uncertainty reasoning capabilities of model-free algorithms. We tested our hypothesis using a state-of-the-art RL architecture for POMDPs, which is widely used and tested in various benchmarks.
>
> Nonetheless, despite it not being the original scope of the paper, we see the value in discussing potential benefits or drawbacks of model-free vs model-based approaches. As the reviewer pointed out, given that Kalmamba is conceptually similar to our approach it is a good candidate for such a comparison.
>
> Both Kalmamba and our approach learn a representation $z_t$ which is then used to train the policy and critics within SAC. The key difference is how such representation is learned. Kalmamba trains the Mamba backbone (and thus, the latent state representation) on an auxiliary variational loss used in generative modelling. Its main goal is to learn a representation $z_t$ conducive to temporally-consistent trajectories of high-dimensional observations. The learned representation is biased towards task-relevant details only indirectly via gradients from a reward prediction model $p(r_t | z_t)$, i.e., the learned representation must lead to accurate reward predictions as well. In summary, Kalmamba trains the latent representation for predicting the dynamics of the underlying MDP (rewards and states). These model-based approaches have the benefit of shaping the latent representation even if the task-relevant signal for learning (i.e., rewards) is weak, which has shown to accelerate learning and improve sample-efficiency. At the same time, the use of auxiliary loss functions which are not optimized jointly with the policy raises concerns regarding objective mismatch [1], where the representation is biased towards objectives that might be either incompatible or misaligned with policy optimization.
>
> (continues in next reply)
>
> [1] Objective Mismatch in Model-based Reinforcement Learning. Lambert et al. 2020

---

> > ### Author Response · Authors · 2024-11-19
> > **Response to Reviewer FQjn (1/2)**
> >
> > In contrast, our model-free approach trains the KF layers only using gradients from the SAC loss function, which directly encodes the RL objective of maximizing returns. The learned representation is explicitly trained for control and not for prediction. This affords some benefits over model-based approaches, namely (1) simplifying the implementation and (2) removing concerns of mismatch between the representation learning and policy optimization. However, auxiliary loss functions (e.g., contrastive, as in CURL, or variational as in Kalmamba) have shown to aid RL training, especially in tasks with complex observation spaces such as images. The rationale behind these performance gains is that the auxiliary objectives provide a meaningful learning signal to shape the latent space, independent of the task-relevant reward signal.
> >
> > ## Broaden the scope of benchmark to include a range on state-space models
> > > The absence of direct comparisons with existing methods, both deterministic SSMs like CURL and probabilistic SSMs like DreamerV3, is a significant limitation.
> >
> > We appreciate the reviewer’s suggestion. As mentioned in the previous point, we are conducting experiments in a set of tasks reported in the Kalmamba paper such that we can compare performance against their reported model-based baselines. We will update the paper with those results and include the relevant discussions.
> >
> > We would like to emphasize that the use of auxiliary loss functions is an orthogonal research direction to ours. It is reasonable to believe that including similar auxiliary losses to our model-free architecture may enhance performance, provided the learned representation is well-aligned with the RL objective.
> >
> > The reduced scope of our experiments enable us to make concrete attributions of performance gains due to the probabilistic inference mechanism of the KF layer within the model-free architecture, which was our main research question and the motivation behind our work. We also acknowledge our methodology narrows down the scope of our results, but it ensures careful validation of our claims; we argue this is an acceptable tradeoff.
> >
> > Considering the challenges of proper benchmarking in RL, we designed our empirical experiments around the principle of fixing as many confounding factors as possible: all our baselines are implemented in the same codebase, sharing all the architectural components except for the history encoders. Moreover, hyperparameters are also shared across baselines and ensure the total parameter counts are on the same orders of magnitude for all the history encoder implementations.
> >
> > ## Test on image-based observations
> > > It is suggested to utilize image-based observations with noisy backgrounds, which better mirror the challenges present in real-world applications, to test the robustness and utility of the proposed model under more demanding conditions.
> >
> > We thank the reviewer for the suggestion to explore noisy, image-based observations. We would like to clarify two key points in this discussion:
> >
> > **Relevance of current results:** we argue that our method and results are highly relevant to the control of physical systems (e.g., in robotics) since (1) it is common to have access to proprioceptive sensors in practice (joint encoders, gyroscopes, etc.), (2) those sensors are noisy, thus the need for probabilistic filtering and (3) inference in KF layers scales adequately for real-time control.
> >
> > **Orthogonal complexity of image-based observations:** we argue that the layer of complexity introduced by image observations is orthogonal to our main research question: do history encoders with probabilistic inference mechanisms bring any benefits to model-free RL under partial observability? In particular, handling images in RL typically involves adding a Convolutional Neural Network (CNN) backbone whose task is to output relevant image features, which we can then feed into the rest of the architecture. The additional learning problem is often addressed with auxiliary loss functions, such as the contrastive learning approach in CURL, which is an interesting orthogonal improvement to our architecture. It is well-studied in the literature that learning directly from images without such auxiliary losses can be fairly sample-inefficient (e.g., the Atari benchmark requiring 200M frames).
> >
> > Nonetheless, we acknowledge the reviewer’s suggestion and have included it as an interesting direction for future work in the revised paper.

---

> > > ### Comment · Reviewer_FQjn · 2024-12-10
> > >
> > > No questions, thanks.

---

### Author Response · Authors · 2024-11-19
**Summary of changes in the paper revision**

We appreciate all of the reviewers’ thoughtful comments. We have now posted replies to all reviewers alongside a revised version of the paper. We are in the process of running additional experiments (to compare against model-based methods, as suggested by Reviewer FQjn) and we will upload a new revision once they are finished. In the current revision, we have made the following changes based on feedback:

- Clarified in the introduction the objective of the paper, the core research question we investigate and a clear statement of our contributions.
- Thoroughly revised the related work section of the paper, which now has a broadened scope and discusses at length tradeoffs and gaps in prior work. (Reviewers CzkF and FQjn).
- Re-wrote the background section to be self-contained by including the relevant simplifications and resulting Kalman filter equations. (Reviewer CzkF)
- Improved clarity and consistency throughout the paper, including the state-space model formulation and the simplifying assumptions. (Reviewer CzkF)

We believe the paper has improved significantly with respect to the initial version and would hope it addresses the major concerns raised by the reviewers.

---

### Author Response · Authors · 2024-11-24
**Comparison to model-based approaches**

We appreciate the reviewers' patience while we finished our new experiments.

In the latest revision of the paper we have included a new Appendix H: DMC Comparison to Model-Based Approaches. In this appendix, we compare and discuss the performance of vSSM and vSSM+KF against state-of-the-art model-based approaches reported in [1]. We observed that vSSM+KF achieves asymptotic performance comparable to these baselines in 3 out of 4 tasks, albeit with lower sample-efficiency. These results demonstrate that performant policies can be trained in these tasks without relying on the representation learning objectives used by the model-based baselines. Furthermore, this comparison opens promising directions for future work, such as improving sample-efficiency while retaining the simplicity of our approach.

We want to emphasize that comparing model-free and model-based methods involves several confounding factors. However, as pointed out by Reviewers CzkF and FQjn, we also agree that such comparisons provide valuable context, broadening the scope of our evaluation and positioning our contributions within the landscape of RL methods.

[1] KalMamba: Towards Efficient Probabilistic State Space Models for RL under Uncertainty, Becker et al. 2024

---

> ### Comment · Action_Editor_aAUe · 2024-11-27
>
> Dear reviewers,
> May I kindly ask you to look at the authors' new experiments and see whether your concerns have been addressed? Please update your review and recommendation.
> Regards,
> AE

---

### Decision · Action_Editor_aAUe · 2025-01-06

**Recommendation:** Accept with minor revision

**Comment:**

Dear authors,

The reviewers are generally satisfies with the changes made in the paper.

However, you mentioned the comparisons to model-based approaches in the rebuttal and made some arguments about the benefits of using model-free RL. However, without verifying theoretical proof or experiments, the arguments are not convincing. Furthermore, planning with a model could help to navigate to deal with the problem of reward sparsity and how you make a trade-off between using model-based and model-free methods according to their pros and cons should be clarified. Therefore, please add those experiments and discussion in the paper to support the arguments.

I will check whether these changes are made in the final version.

Thanks,
AE

**Audience:**

Yes

**Claims And Evidence:**

Yes

---

> ### Author Response · Authors · 2025-02-18
> **Response to AE**
>
> Dear AE,
>
> We appreciate the feedback received throughout the review process and are pleased by the positive effect it had on our manuscript.
>
> In Appendix H: DMC Comparison to Model-Based Approaches, we compare and discuss the performance of vSSM and vSSM+KF against state-of-the-art model-based approaches. We refer to this appendix in the main text at the end of Section 5.3 of the camera-ready version, alongside a short discussion of its main insight.